# Semaphorin 5A inhibits synaptogenesis in early postnatal- and adult-born hippocampal dentate granule cells

**Yuntao Duan[1†], Shih-Hsiu Wang[2,3†‡], Juan Song[3,4§], Yevgeniya Mironova[1], Guo-li Ming[3,4], Alex L Kolodkin[2,3]\*, Roman J Giger[1,5]\***

[1]Department of Cell and Developmental Biology, University of Michigan School of Medicine, Ann Arbor, United States; [2]Howard Hughes Medical Institute, Johns Hopkins University School of Medicine, Baltimore, United States; [3]Solomon H Snyder Department of Neuroscience, Johns Hopkins University School of Medicine, Baltimore, United States; [4]Department of Neurology, Johns Hopkins University School of Medicine, Baltimore, United States; [5]Department of Neurology, University of Michigan School of Medicine, Ann Arbor, United States

**\*For correspondence:**
kolodkin@jhmi.edu (ALK);
rgiger@umich.edu (RJG)

[†]These authors contributed equally to this work

**Present address:** [‡]Department of Pathology and Cell Biology, Columbia University College of Physicians and Surgeons, New York, United States; [§]Department of Pharmacology and Neuroscience Center, University of North Carolina, Chapel Hill, United States

**Competing interests:** The authors declare that no competing interests exist.

**Reviewing editor**: Freda Miller, The Hospital for Sick Children Research Institute, University of Toronto, Canada

**Abstract** Human *SEMAPHORIN 5A (SEMA5A)* is an autism susceptibility gene; however, its function in brain development is unknown. In this study, we show that mouse *Sema5A* negatively regulates synaptogenesis in early, developmentally born, hippocampal dentate granule cells (GCs). Sema5A is strongly expressed by GCs and regulates dendritic spine density in a cell-autonomous manner. In the adult mouse brain, newly born *Sema5A*$^{-/-}$ GCs show an increase in dendritic spine density and increased AMPA-type synaptic responses. Sema5A signals through PlexinA2 co-expressed by GCs, and the PlexinA2-RasGAP activity is necessary to suppress spinogenesis. Like *Sema5A*$^{-/-}$ mutants, *PlexinA2*$^{-/-}$ mice show an increase in GC glutamatergic synapses, and we show that *Sema5A* and *PlexinA2* genetically interact with respect to GC spine phenotypes. *Sema5A*$^{-/-}$ mice display deficits in social interaction, a hallmark of autism-spectrum-disorders. These experiments identify novel intra-dendritic Sema5A/PlexinA2 interactions that inhibit excitatory synapse formation in developmentally born and adult-born GCs, and they provide support for SEMA5A contributions to autism-spectrum-disorders.

## Introduction

In the mammalian CNS, most excitatory neurotransmission takes place at spiny synapses, and our understanding of the mechanisms that control the density and strength of excitatory glutamatergic synapses remains incomplete. Many synaptogenic molecules have been identified that can induce pre- or postsynaptic differentiation (*Allen and Barres, 2005*; *Shen and Cowan, 2010*; *Siddiqui and Craig, 2011*; *de Wit et al., 2011*). Much less is known, however, about molecular players that prevent formation of supernumerary spine synapses (*Chung and Barres, 2012*; *Mironova and Giger, 2013*). Precise regulation of CNS synapse density is critical for proper brain function and mental health, and imbalances in excitatory and inhibitory synaptic transmission are associated with neurodevelopmental disorders such as autism-spectrum disorders (ASD) and schizophrenia (*Penzes et al., 2011*).

The dentate gyrus (DG) is one of two neurogenic areas in the adult mammalian brain (*Altman and Bayer, 1990*; *Ming and Song, 2011*). Proper insertion of adult-born granule cells (GCs) into a pre-existing synaptic network offers a unique opportunity to study mechanisms governing axon guidance, dendrite elaboration and formation of synaptic contacts that contribute to neuronal plasticity in mature nervous tissue. Despite recent progress (*Tran et al., 2009*; *Siddiqui et al., 2013*; *de Wit et al., 2013*), our

**eLife digest** Neurons communicate with one another at specialized junctions called synapses. There are two types of synapses, called excitatory synapses and inhibitory synapses, and the density and strength of both are tightly regulated because small deviations from the normal density and/or strength may lead to illness. For example, an excess of excitatory synapses has been observed in patients who have autism spectrum disorders and exhibit difficulties in social interaction.

The gene that codes for a protein called SEMA5A has been identified as an autism susceptibility gene in humans. SEMA5A is a transmembrane protein that regulates the development of connections between neurons, but it is not known how mutations in the gene for SEMA5A might lead to brain illnesses such as autism spectrum disorders.

Now, Duan et al. report that Sema5A selectively inhibits the formation of excitatory synapses in neurons called dentate granule cells in mice. Moreover, Sema5A reduces the frequency and amplitude of the signals that pass through excitatory synapses in certain granule cells. Sema5A directly binds to the receptor PlexA2, a protein that is involved in controlling the density of synapses. In behavioral studies, Sema5A mutant mice displayed altered patterns of social interaction compared to control animals, being less willing to interact with unfamiliar mice.

The presence of increased numbers of excitatory synapses in the brains of *Sema5A* mutant mice implies that expression of the *Sema5A* gene normally prevents the formation of too many excitatory synapses. The fact that these animals also show altered social behavior suggests that an excess of synapses—whether as a result of increased synapse formation and/or reduced synapse elimination—can lead to changes in brain circuitry that give rise to patterns of behavior that are characteristic of autism spectrum disorders.

understanding of the molecular programs that regulate GC synaptogenesis is incomplete, including understanding the extent to which the same molecular cues can direct developmentally born and adult-born GCs to establish proper synaptic connectivity (*Toni and Sultan, 2011*; *Kim et al., 2012*; *Schnell et al., 2012*).

One family of extracellular cues known to regulate the morphology of developmentally born GCs is the semaphorins. Semaphorin 6A (Sema6A) and Sema6B inhibit growth of GC axons in vitro and are necessary for laminar targeting of mossy fiber (MF) projections in the CA3 subregion early during postnatal development (*Suto et al., 2007*; *Tawarayama et al., 2010*). Further, dendritic elaboration, spine density, and synaptic transmission in GCs are regulated, at least in part, by secreted class 3 semaphorins (*Sahay et al., 2005*; *Tran et al., 2009*; *Ng et al., 2013*).

Sema5A and Sema5B are two closely related transmembrane proteins with an extracellular sema-domain followed by a cluster of seven type-1 thrombospondin repeats (TSRs). The cytoplasmic domains of Sema5A and Sema5B are ~85 amino acid residues in length and harbor no obvious signaling motifs (*Adams et al., 1996*; *Tran et al., 2009*). In vitro, Sema5A functions as an axon guidance molecule for various subclasses of primary neurons (*Goldberg et al., 2004*; *Kantor et al., 2004*), and Sema5B regulates the development of synaptic contacts in hippocampal neurons in vitro (*O'Connor et al., 2009*). In addition, *Sema5A* and *Sema5B* together insure proper stratification of murine retinal neuron projections, utilizing both *Plxna1* and *Plxna3* as receptors (*Matsuoka et al., 2011*).

Genome wide-association studies identify *SEMA5A* as an ASD susceptibility gene (*Weiss et al., 2009*). However, the role of Sema5A in mammalian brain development and physiology has not been addressed in vivo. Here, we find that *Sema5A,* but not *Sema5B,* negatively regulates synaptic density in both developmentally born and adult-born dentate GCs. PlexinA2 is a novel receptor for Sema5A, and we show that loss of *Sema5A* leads to increased excitatory synaptic transmission and ASD-like behavioral phenotypes.

## Results

### Class 5 semaphorins are expressed in dentate GCs and enriched in the post-synaptic density

*Sema5A* and *Sema5B* are expressed in the rodent hippocampus (*Simmons et al., 1998*; *O'Connor et al., 2009*). To augment these data, we conducted a detailed analysis of *Sema5A* expression in the

postnatal hippocampus and entorhinal cortex (EC) using a *Sema5A* reporter mouse expressing nuclear *lacZ* (*Gunn et al., 2011*). In the P18 and P30 hippocampus, strong β-gal activity is observed in the granule cell layer (GCL), the hilus and in CA3 pyramidal neurons. Moderate-to-strong β-gal activity is also observed in deep EC layers (*Figure 1A*), however more superficial EC layers, including layer II/III neurons that give rise to the perforant path, do not show *Sema5A* promoter activity. In situ hybridization for *Sema5A* revealed a similar expression pattern (*Figure 1B*). *Sema5B* expression in the postnatal hippocampus is largely confined to the DG subgranular zone (SGZ). Subcellular fractionation of hippocampal tissue revealed that Sema5A protein is present in synaptic and extra-synaptic membrane fractions and is enriched in detergent-resistant post-synaptic density (PSD) fractions. Sema5B is enriched postsynaptically and only found in the PSDIII fraction (*Figure 1C*).

Analysis of *Sema5A* and *Sema5B* null mutants (*Matsuoka et al., 2011*) at the macroscopic level revealed no obvious defects in the hippocampal formation (*Figure 1—figure supplement 1*). GCs in compound heterozygotes and null mutants were analyzed using anti-doublecortin, anti-calretinin, and anti-calbindin labeling. No defects were identified using these markers, suggesting that *Sema5A* and *Sema5B* are dispensable for GC maturation and for general patterning of GC axonal and dendritic processes in vivo.

## Loss of *Sema5A*, but not *Sema5B*, increases GC dendritic spine density

We next examined dendritic spine distribution in P30–33 hippocampal neurons in *Sema5*;*Thy1-GFPm* reporter mice, analyzing dendritic spines in the middle and outer thirds of supra-pyramidal GC dendrites within the rostral hippocampus (*Figure 1D*). We also quantified spine density along apical dendrites of hippocampal CA1 pyramidal neurons. Spine density in WT GCs (1.67 ± 0.01 spines/μm), *Sema5A*$^{+/-}$;*Sema5B*$^{+/-}$ (1.68 ± 0.05 spines/μm), and *Sema5A*$^{+/-}$;*Sema5B*$^{-/-}$ mice (1.66 ± 0.05 spines/μm) is similar. However, GC spine density in *Sema5A*$^{-/-}$;*Sema5B*$^{+/-}$ mice (2.05 ± 0.05 spines/μm) is increased by 23 ± 3% compared to WT, and no further increase is observed in *Sema5A*$^{-/-}$;*Sema5B*$^{-/-}$ double-mutants (2.06 ± 0.08 spines/μm) (*Figure 1E,F*, *Table 1*). Spine density along CA1 pyramidal neuron primary and secondary apical dendrites is not altered by the loss of *Sema5s* (*Figure 1—figure supplement 2* and *Table 1*). To conditionally ablate *Sema5A* in P15 GCs, neuron-specific recombination in *Sema5A*$^{flox/-}$ mice was achieved by stereotaxic injection of a Cre recombinase-expressing lentiviral vector. Similar to *Sema5A* germline null mice, conditional *Sema5A* ablation in GCs leads to increased spine density (*Figures 1G-I*). The vast majority of excitatory inputs onto GCs and pyramidal neurons form on dendritic spines (*Harris and Kater, 1994*; *Trommald and Hulleberg, 1997*). Our results suggest that Sema5A inhibits the formation of excitatory synapses on GCs. To assess the density of excitatory and inhibitory GC synapses, *Sema5A* WT and null neurons were cultured and labeled with anti-PSD95 and anti-gephyrin (*Figure 1J–L*). We found that *Sema5A*$^{-/-}$ GCs exhibit a selective increase in excitatory, but not inhibitory, synapses.

## Loss of *Sema5A* negatively regulates dendritic spine density and alters synaptic transmission in adult-born GCs

We next used retrovirus-mediated birth-dating to permanently label newly born DG neurons in the adult mouse (*Ge et al., 2006*). We injected an onco-retroviral vector harboring a GFP transgene into the hilus of 6- to 8-week-old mice, labeling proliferating neural progenitors (*Figure 2A–F*). GC dendritic spine density was assessed 19–21 days after injection, the peak time for the formation of glutamatergic synaptic inputs onto adult-born GCs (*Toni et al., 2007*). At this developmental time point, adult-born GCs have spiny dendrites that reach the outer ML and begin acquiring excitatory glutamatergic input from the EC (*Esposito et al., 2005*; *Zhao et al., 2006*). In WT mice, the dendritic spine density of adult-born WT GCs (0.96 ± 0.07 spines/μm) was lower than that of fully mature WT GCs (1.67 ± 0.01 spines/μm). However, in *Sema5A*$^{-/-}$ mice, spine density of adult-born GCs was significantly increased (1.65 ± 0.08 spines/μm) and comparable to mature WT GCs (1.67 ± 0.01 spines/μm) (*Figure 2G*). This shows that *Sema5A* negatively regulates dendritic spine development in adult-born GCs in vivo.

Glutamatergic inputs onto adult-born GCs appear between 14 and 28 days post mitosis (*Esposito et al., 2005*; *Ge et al., 2006*), and we next performed whole-cell patch clamp recordings in acute hippocampal slices prepared from *Sema5A*$^{-/-}$ mice and control littermates (*Figure 2H–K*). To identify adult-born GCs, and also to ensure that recordings were from neurons at the same postmitotic stage, we combined electrophysiological recordings with retrovirus-mediated birth-dating. Analysis of the frequency and amplitude of miniature excitatory postsynaptic currents (mEPSCs) in 19- to 21-day

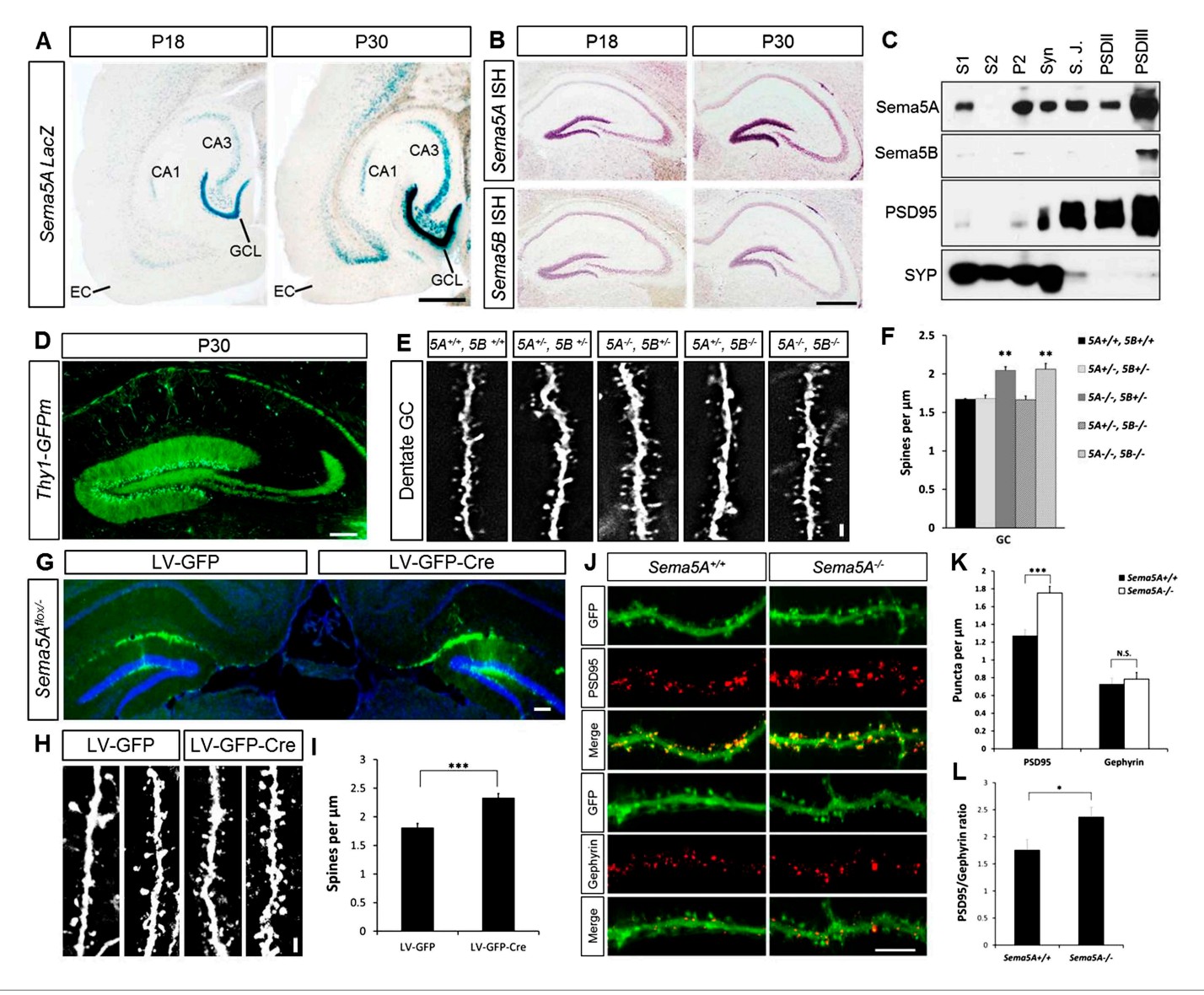

**Figure 1**. *Sema5A*, but not *Sema5B*, negatively regulates dendritic spine density of hippocampal GCs in vivo. (**A**) Horizontal sections through the hippocampus of *Sema5A^LacZ/+* brains show robust β-gal activity in the GCL at P18 and P30. Weaker labeling is observed in the pyramidal cell layer of CA3 and a small segment of CA1. In the EC, labeling is confined to deep cortical layers. (**B**) In situ hybridization of P18 and P30 coronal sections with probes specific for *Sema5A* and *Sema5B*. (**C**) Western blot analysis of synaptic density fractions prepared from P18 mouse hippocampus. S1, homogenate; S2, cytosolic fraction; P2, membrane fraction; Syn, synaptosomal fraction; S.J., synaptic junction, and PSD purified postsynaptic fractions. Anti-PSD95 and anti-Synaptophysin (SYP) are shown as post- and pre-synaptic markers. (**D**) Coronal section of the P33 *Thy1-GFPm* hippocampus showing labeling of GCs. (**E**) Representative images of GFP-positive dendrites of GCs of WT (*5A^+/+, 5B^+/+*), *Sema5A^+/−*, *Sema5B^+/−* (*5A^+/−, 5B^+/−*); *Sema5A^−/−*, *Sema5B^+/−* (*5A^−/−, 5B^+/−*); *Sema5A^+/−*, *Sema5B^−/−* (*5A^+/−, 5B^−/−*) and double mutant (*5A^−/−, 5B^−/−*) mice. (**F**) Quantification of dendritic spine density of GCs shown in **E**. Values are represented as mean ± SEM from 3 to 4 mice per genotype (for details on spine quantification see *Table 1*). \*\*indicates p < 0.01, two-tailed unpaired Student's *t* test. (**G**) Representative image of the DG following stereotaxic injection of LV-GFP (left side) or LV-syn-GFP-IRES-Cre (right side) viral vector into *Sema5A^flox/−* mice. (**H**) High magnification images of LV transduced GC dendrites. (**I**) Quantification of spine density of GFP^+ GC dendrites. Values are represented as mean ± SEM from three independent mice per condition. (**J**) Cultured mouse hippocampal neurons at DIV21 obtained from *Sema5A^+/+* and *Sema5A^−/−* pups. Cultures were transfected at DIV4 with a GFP expression construct and fixed at DIV21. GCs were identified by anti-Prox1 labeling (data not shown). Cultures were stained with anti-PSD95 or anti-gephyrin to identify excitatory synapses confined to dendritic spines, and inhibitory synapses confined to the dendritic shaft of GCs. (**K**) Quantification of PSD95 positive puncta reveals a significant increase in *Sema5A^−/−* GCs. No significant (N.S.) difference in gephyrin positive puncta was observed between *Sema5A^+/+* and *Sema5A^−/−* GCs. (**L**) The ratio of excitatory/inhibitory synapses is significantly increased in *Sema5A^−/−* GCs. Number of neurons quantified: n = 32–34 neurons per condition from three mice per genotype.

*Figure 1. Continued on next page*

*Figure 1. Continued*

Values are represented as mean ± SEM. \*\*\*p < 0.001; \*p < 0.05 two-tailed unpaired Student's *t* test. Scale bars, **A** and **B** = 500 μm, **D** and **G** = 200 μm, **J** = 10 μm, **E** and **H** = 1 μm.

The following figure supplements are available for figure 1:

**Figure supplement 1**. Normal maturation and patterning of dentate GCs in *Sema5A*$^{-/-}$ ; *Sema5B*$^{-/-}$ double mutants.

**Figure supplement 2**. *Sema5* mutant mice do not show altered spine density in CA1 pyramidal neurons.

postmitotic GCs of 9- to 11-week-old mice revealed a significant increase in mEPSC frequency and amplitude in *Sema5A*$^{-/-}$ hippocampal slices compared to WT slices (**Figure 2I–K**). Thus, in *Sema5A*$^{-/-}$ mice, increased GC spine density (**Figure 1E**) is correlated with an increase in mEPSC frequency, suggesting that supernumerary spines receive presynaptic input and are electrically active. Moreover, since these mEPSCs are fully blocked in the presence of CNQX, this suggests that loss of *Sema5A* leads to increased AMPA receptor responses.

## Sema5A cell-autonomously inhibits GC dendritic spine density

The expression of *Sema5A* in dentate GCs, but not EC superficial layers, coupled with our observation that conditional ablation of *Sema5A* in the DGL increases dendritic spine density, strongly suggest that Sema5A functions in a GC-autonomous manner. To address whether Sema5A influences spine density through interactions in *trans* between neighboring GCs or in *cis* within the same cell, we employed primary hippocampal neurons stained for the GC marker prospero homeobox protein 1 (prox1) (**Elliott et al., 2001**) (**Figure 3**). At DIV21, *Sema5A*$^{-/-}$ GCs show a significant (33.5%) increase in spine density (2.23 ± 0.08 spines/μm) as compared to WT controls (1.67 ± 0.04 spines/μm). To ask whether the supernumerary GC spine phenotype is rescued by recombinant Sema5A, neuronal cultures were transfected at DIV4 and analyzed at DIV21 (**Figure 3A**). Because only a small fraction (~1%) of the neurons in culture is transfected, and since spine density was assessed in isolated, prox1$^+$, neurons whose processes did not overlap with any other *Sema5A*-transfected neurons, *trans* intercellular Sema5A interactions do not figure in this analysis. Recombinant Sema5A was present on dendrites and cell bodies (**Figure 3—figure supplement 1A,B**) and decreased GC spine density by 34% in *Sema5A*$^{-/-}$ (1.10 ± 0.04 spines/μm) and by 29% in *Sema5A*$^{+/+}$ (1.18 ± 0.04 spines/μm) hippocampal cultures (**Figure 3A,B**). This suggests that a Sema5A *cis* interaction cell-autonomously constrains the formation of supernumerary GC spines. In these same cultures, expression of recombinant Sema5A in pyramidal neurons leads to a reduction in spine density (**Figure 3—figure supplement 2**). To examine whether or not Sema5A serves as a receptor (**Pasterkamp, 2012**), we expressed a Sema5A deletion mutant lacking its cytoplasmic domain (S5A$^{\Delta cyto}$) and asked if this altered Sema5A protein was sufficient to rescue the *Sema5A*$^{-/-}$ GC spine phenotype. Similar to full-length Sema5A (1.10 ± 0.04 spines/μm), S5A$^{\Delta cyto}$ rescues the *Sema5A*$^{-/-}$ spine phenotype (1.09 ± 0.03 spines/μm) (**Figure 3A,B**). We next generated Sema5A deletion constructs lacking either the sema-domain (S5A$^{\Delta Sema}$) or the 7 TSRs (S5A$^{\Delta TSR}$), confirmed that they were appropriately expressed (data not shown), and found that S5A$^{\Delta Sema}$ (1.94 ± 0.05 spines/μm), but not S5A$^{\Delta TSR}$ (1.11 ± 0.05 spines/μm), fails to rescue the *Sema5A*$^{-/-}$ GC spine phenotype (**Figure 3A,B**). These results show that the sema-domain of Sema5A is necessary and sufficient to regulate GC dendritic spine density. Further, they suggest that Sema5A functions as a ligand or co-receptor that interacts in *cis* with a transmembrane-spanning signal transducing receptor to constrain GC spine formation. Independent evidence that Sema5A functions as a ligand stems from experiments in which we bath applied soluble Sema5A-Fc fusion protein to DIV21 GCs. Sema5A-Fc, but not control IgG protein, results in a rapid loss of dendritic spines (**Figure 3C,D**).

## Sema5A binds directly to PlexA1 and PlexA2 with high affinity

We next assessed Sema5A interactions with PlexA family members since class 5 semas are known to utilize class A plexins as receptors in vivo (**Matsuoka et al., 2011**). Sema5A binds directly and with low-nanomolar affinity to PlexinA1 and PlexinA2 ectopically expressed in COS7 cells, and the sema-domain of Sema5A binds to the sema-domains of PlexA1 and PlexA2 (**Figure 4A–C**, **Figure 4—figure supplement 1**). We next assessed hippocampal neurons from *Plxna1*, *Plxna2*, and *Plxna3* homozygous mutant pups for their ability to support Sema5A-Fc binding. Sema5A-Fc bound strongly to WT and

**Table 1.** Quantification of dendritic spine density in the hippocampus of *Thy1-eGFP* WT, *Sema5* and *Plxna* mice

| | | | Dentate GC dendrite | | | CA1 primary dendrite | | | | CA1 secondary dendrite | | | |
| --- | --- | --- | --- | --- | --- | --- | --- | --- | --- | --- | --- | --- | --- |
| Genotype | Age | N | p Value (vs wt) | Spine density (mean ± SEM) | Distance from soma | n | p Value (vs wt) | Spine density (mean ± SEM) | Distance from soma | n | p Value (vs wt) | Spine density (mean ± SEM) | Distance from soma |
| WT | P30–33 | 4 animals (74 neurons) | | 1.67 ± 0.01 | 50–100 μm | 3 animals (26 neurons) | | 2.76 ± 0.07 | 50–100 μm | 3 animals (15 neurons) | | 2.93 ± 0.02 | 50–100 μm |
| *Sema5A⁺/⁻; Sema5B⁺/⁻* | P30–33 | 3 animals (41 neurons) | 0.8795 | 1.68 ± 0.05 | 50–100 μm | 3 animals (24 neurons) | 0.6549 | 2.79 ± 0.019 | 50–100 μm | 3 animals (24 neurons) | 0.6333 | 2.99 ± 0.09 | 50–100 μm |
| *Sema5A⁻/⁻; Sema5B⁺/⁻* | P30–33 | 4 animals (65 neurons) | **0.0012** | 2.05 ± 0.05 | 50–100 μm | 3 animals (27 neurons) | 0.5122 | 2.81 ± 0.03 | 50–100 μm | 3 animals (21 neurons) | 0.618 | 2.99 ± 0.09 | 50–100 μm |
| *Sema5A⁺/⁻; Sema5B⁻/⁻* | P30–33 | 3 animals (46 neurons) | 0.8844 | 1.66 ± 0.05 | 50–100 μm | 3 animals (26 neurons) | 0.9797 | 2.76 ± 0.15 | 50–100 μm | 3 animals (21 neurons) | 0.9661 | 2.94 ± 0.02 | 50–100 μm |
| *Sema5A⁻/⁻; Sema5B⁻/⁻* | P30–33 | 3 animals (30 neurons) | **0.0066** | 2.06 ± 0.08 | 50–100 μm | 3 animals (27 neurons) | 0.5389 | 2.81 ± 0.02 | 50–100 μm | 3 animals (23 neurons) | 0.4006 | 2.98 ± 0.03 | 50–100 μm |
| *Plxna1⁻/⁻* | P30–33 | 3 animals (56 neurons) | 0.154 | 1.79 ± 0.05 | 50–100 μm | | | | | | | | |
| *Plxna2⁻/⁻* | P30–33 | 3 animals (54 neurons) | **<0.0001** | 2.24 ± 0.07 | 50–100 μm | | | | | | | | |
| *Plxna2⁺/⁻* | P30–33 | 4 animals (74 neurons) | 0.0359 | 1.86 ± 0.05 | 50–100 μm | | | | | | | | |
| *Sema5A⁺/⁻; Plxna2⁺/⁻* | P30–33 | 4 animals (72 neurons) | **<0.0001** | 2.39 ± 0.05 | 50–100 μm | | | | | | | | |

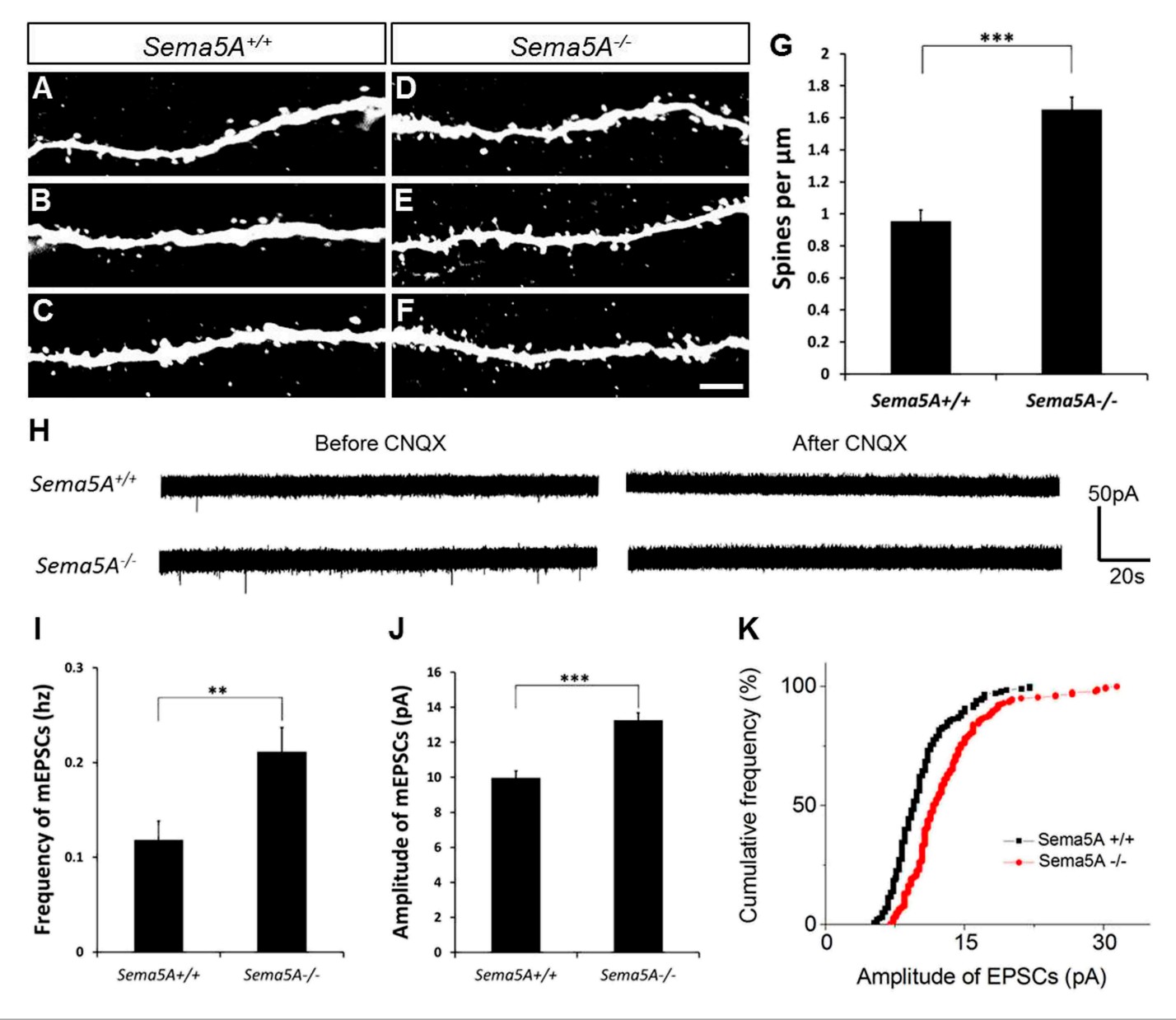

**Figure 2**. In adult-born GCs, *Sema5A* negatively regulates dendritic spine density and affects their electrophysiological properties. (**A–F**) Representative images of dendritic segments of adult-born, 19- to 21-day-old retrovirally-labeled GCs in 3-month-old *Sema5A⁺/⁺* and *Sema5A⁻/⁻* mice. (**G**) Quantification of dendritic spine density in adult-born GCs of *Sema5A⁻/⁻* (n = 34 neurons) and *Sema5A⁺/⁺* (n = 35 neurons) mice. Error bars represent ± SEM, n = 3 independent animals per genotype. ***indicates p < 0.0001, two-tailed unpaired Student's *t* test. Scale bar = 5 μm. (**H–K**) Recordings of spontaneous mEPSCs from retrovirally-labeled adult-born GCs at 19–21 days postmitosis. (**H**) Shown are sample voltage-clamp whole-cell recording traces (V_m = −65 mV) from *Sema5A⁺/⁺* (top) and *Sema5A⁻/⁻* (bottom) animals before and after bath application of CNQX (20 μM). (**I**) Quantification of AMPA-type mEPSC mean frequency and (**J**) mean amplitude. (**K**) Relative cumulative frequency distribution of mEPSC amplitudes in *Sema5A⁺/⁺* (black) and *Sema5A⁻/⁻* (red) GCs. The distribution of mEPSC amplitudes in mutants is significantly shifted to the right, indicating an increase in the proportion of larger amplitude events. Recordings were performed in the presence of TTX (1 μM) and bicuculline (10 μM). Values are represented as mean ± SEM. *Sema5A⁺/⁺*, n = 13 cells/6 mice and *Sema5A⁻/⁻*, n = 13 cells/6 mice. **p < 0.01, ***p < 0.001, Student's *t* test.

*Plxna3⁻/⁻* hippocampal cultures, however, binding to *Plxna1⁻/⁻* and *Plxna2⁻/⁻* neurons was greatly reduced, and virtually no binding was observed to *Plxna1⁻/⁻;Plxna2⁻/⁻* double mutant neurons. In the same assay, binding of an unrelated Fc-fusion protein, NgR^OMNI-Fc (***Robak et al., 2009***), is not influenced by the loss of *Plxna* family members (***Figure 4D–F***). Therefore, PlexA1 and PlexA2 are the main

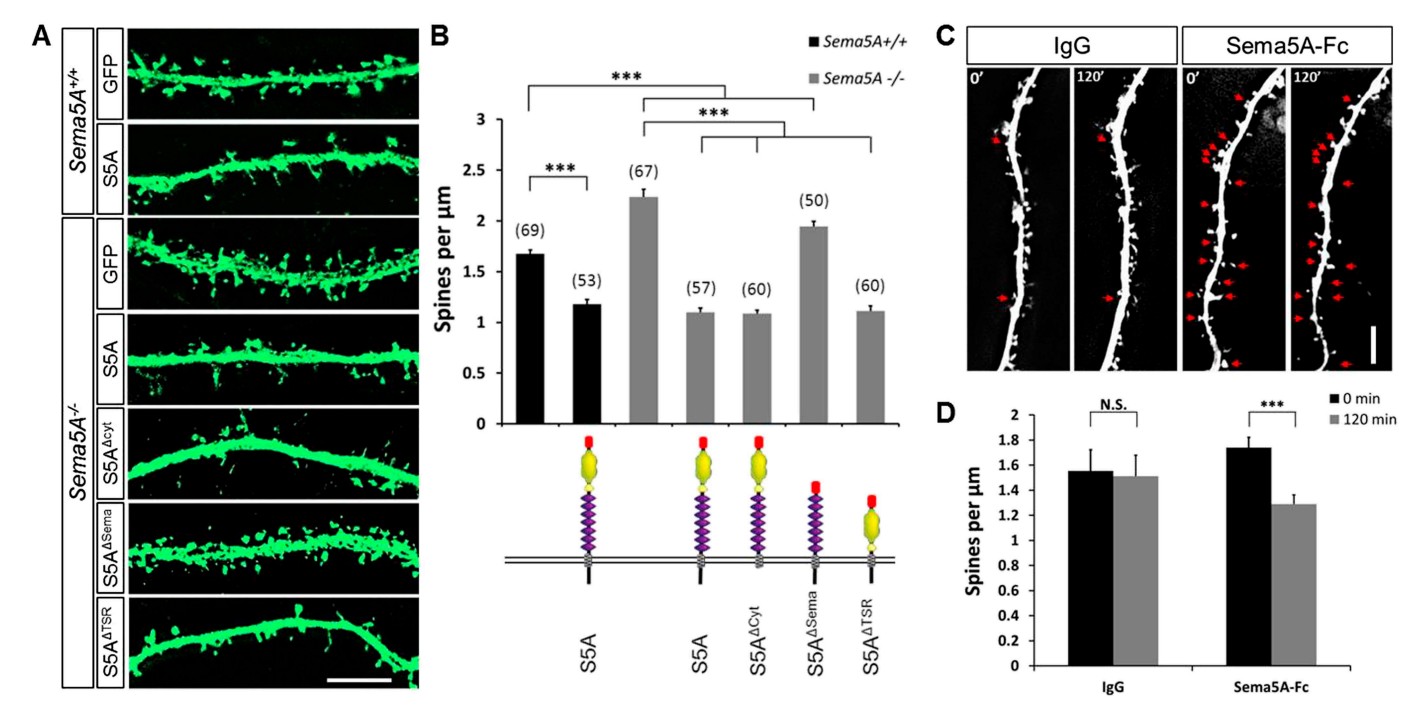

**Figure 3**. *Sema5A* inhibits dendritic spine density in primary dentate GCs. (**A**) High power deconvolution images of dendritic segments of GCs from WT (*Sema5A⁺/⁺*) and mutant (*Sema5A⁻/⁻*) hippocampal cultures transfected either with eGFP only, or eGFP plus full-length Sema5A (S5A), the Sema5A deletion constructs S5A^ΔCyt, S5A^ΔSema, or S5A^ΔTSR plasmid DNA. (**B**) Quantification of dendritic spine density of DIV21 prox1⁺ GCs. Numbers in brackets indicate the number of cells analyzed per condition. (**C**) Time-lapse study of Sema5A-Fc induced GC dendritic spine loss. DIV21 dentate cultures were treated for 120 min with Sema5A-Fc or control IgG. (**D**) Quantification of dendritic spine density in prox1⁺ GCs. Values are represented as mean ± SEM from 3 to 4 cultures, established from 3 to 4 different animals for each condition. ***$p < 0.001$, two-tailed unpaired Student's *t* test. Scale bars in **A** and **C** = 10 μm.

The following figure supplements are available for figure 3:

**Figure supplement 1**. Sema5A is localized to neurites in dissociated hippocampal neurons.

**Figure supplement 2**. Ectopic expression of Sema5A in hippocampal pyramidal neurons decreases dendritic spine density.

Sema5A-Fc binding partners in cultured hippocampal neurons. Conversely, PlexA2^sema-Fc binds to neurites of WT hippocampal neurons, but not to *Sema5A⁻/⁻* neurons, showing that endogenous Sema5A is localized at the neuronal cell surface and that it is also present on neurites (***Figure 4—figure supplement 2***).

## PlexA2, but not PlexA1, is strongly expressed in the postnatal dentate GCL

We next explored which PlexA family members serve as functional receptors for Sema5A in dentate GCs. We analyzed *Plxna1*, *Plxna2,* and *Plxna3* gene expression in the hippocampus by in situ hybridization at P7 and P30 (***Figure 5A***) since the peak period of synaptogenesis between EC and GC neurons occurs between postnatal week two and four (***Esposito et al., 2005***). *Plxna2*, but not *Plxna1*, transcripts are abundantly found in the DG before the onset (P7) and toward the end (P30) of early postnatal GC synaptogenesis (***Figure 5A***). *Plxna3* expression in the DG is quite robust at P7, and at P30 it is largely confined to the SGZ, suggesting higher expression in immature GCs (***Figure 5A***). In hippocampal lysates, PlexA2 protein is most abundant during early postnatal synaptogenesis and continues to be present, but at a lower level, in adulthood (***Figure 5B***).

PlexA1, PlexA2, and PlexA3 are all found in hippocampal homogenates; however, the subcellular distribution patterns of these proteins in P23–P26 synaptic density fractions are quite distinct

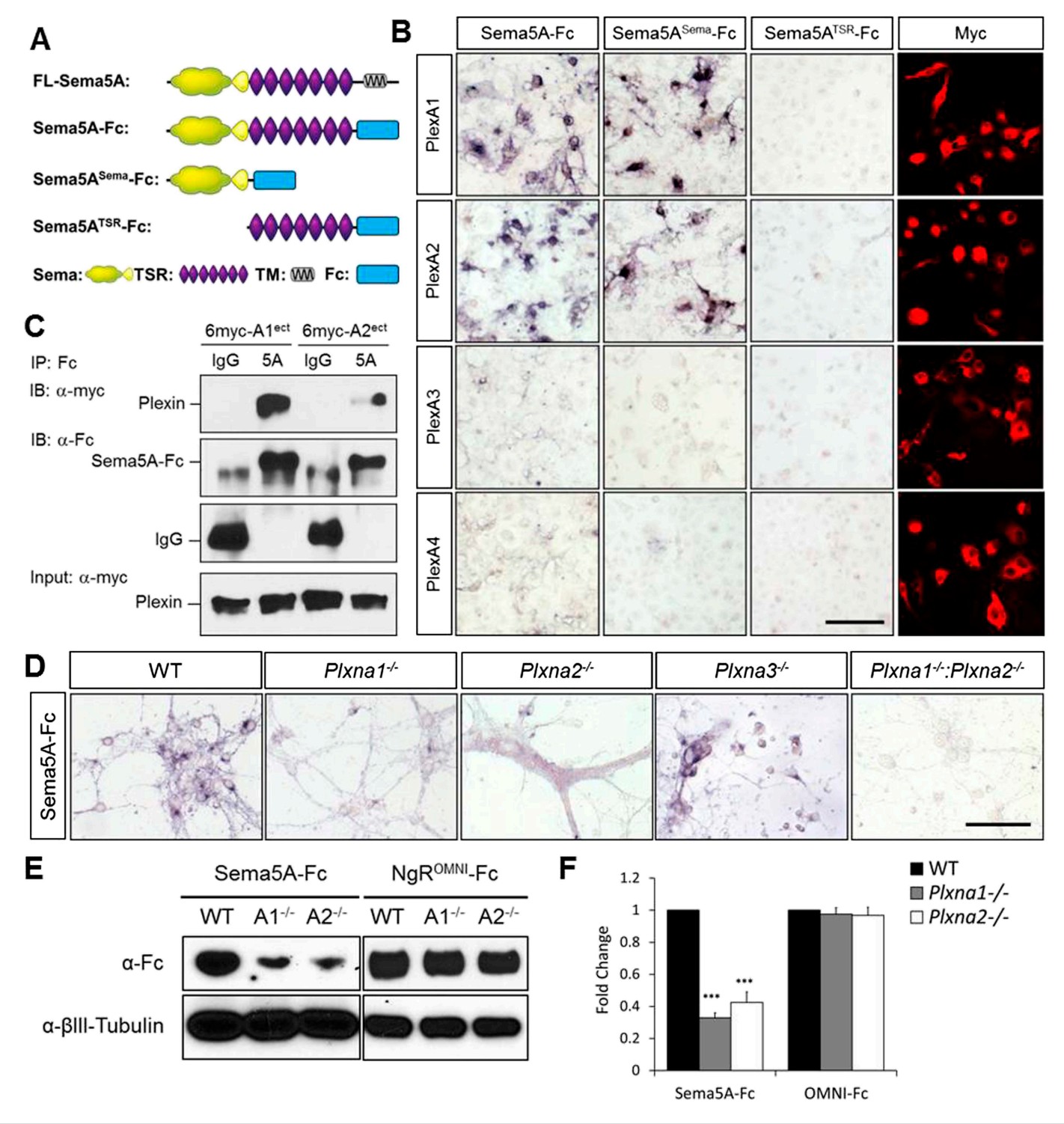

**Figure 4**. Direct binding of Sema5A to PlexA1 and PlexA2. (**A**) Schematic diagrams of Sema5A domain deletion constructs used for binding studies. (**B**) Binding of different Sema5A-Fc fusion proteins to COS7 cells transiently expressing myc-tagged PlexA1, PlexA2, PlexA3, or PlexA4. Sema5A-Fc and Sema5A^Sema-Fc bind selectively to PlexA1 and PlexA2. PlexA expression was confirmed by anti-myc immunostaining. (**C**) Co-precipitation of purified ectodomain of Sema5A (Sema5A-Fc) with the ectodomains of PlexA1 (6x-myc-PlexA1^ect) or PlexA2 (6x-myc-PlexA2^ect). Immunoprecipitation (IP) with anti-Fc shows that Sema5A-Fc (5A), but not control IgG, forms a complex with the ectodomains of PlexA1 and PlexA2. The precipitated plexin fragments can be detected by anti-myc immunoblotting (top blot). Immunoblotting with an antibody specific for the Fc fragment detects Sema5A-Fc and control
*Figure 4. Continued on next page*

*Figure 4. Continued*

IgG in the precipitate (second blot from top). Note that the higher molecular weight band is Sema5A-Fc and the lower molecular weight band is intact (not fully reduced ~150-kDa) IgG. Anti-IgG immunoblotting shows that control IgG at 50-kDa was successfully pulled down with protein A/G agarose-beads (third blot from top). To demonstrate that equal amounts of plexinA ectodomain were used for the IP experiment, an aliquot of the input was probed with anti-myc (blot at bottom). (**D**) Binding of Sema5A-Fc to primary hippocampal neurons established from WT, *Plxna1*, *Plxna2* and *Plxna3* single null mice as well as *Plxna1; Plxna2* double null mice. (**E**) Hippocampal neurons from WT, *Plxna1−/−*, and *Plxna2−/−* mice were incubated with either Sema5A-Fc (50 nM) or NgR$^{OMNI}$-Fc (50 nM) for 1 hr. Cells were rinsed extensively, lysed, and subjected to WB analysis. Sema5A-Fc binding to *Plxna1−/−* and *Plxna2−/−* neurons is reduced compared to WT neurons. NgR$^{OMNI}$-Fc binding was strong to hippocampal neurons and independent of the Plxna genotype. Anti-βIII tubulin in lysates is shown as a loading control. (**F**) Quantification of Western blot signals from three independent experiments. Scale bars in **B** and **D** = 50 μm.

The following figure supplements are available for figure 4:

**Figure supplement 1**. Molecular basis of PlexA1 and PlexA2 binding to Sema5A.

**Figure supplement 2**. Sema5A can be detected on the neuronal surface and is localized to neurites.

(*Figure 5C*). PlexA1, but not PlexA2 or PlexA3, is enriched in the synaptic junction (SJ) and detergent-resistant postsynaptic density (PSD) fractions. PlexA2 and PlexA3 are present in synaptosomes (syn) but are not enriched in the SJ or PSD fractions. This suggests PlexA2 and PlexA3 are preferentially localized at extra-synaptic sites, including the dendritic shaft and spine neck, while PlexA1 is enriched in the PSD near the active zone. Therefore, Sema5A and PlexinA2 are both expressed by GCs and show overlapping distributions in synaptosomal density fractions. *Plxna2* in situ hybridization on adult *Sema5A^{lacZ}* reporter brains followed by anti-β-galactosidase staining revealed double-labeled cells (*Figure 5—figure supplement 1*). This suggests that Sema5A and PlexA2 are co-expressed by GCs.

## PlexA2 is necessary for Sema5A-mediated regulation of dentate GC dendritic spine density

To investigate whether PlexA2 is a functional Sema5A receptor, we analyzed spine density in cultured *Plxna2−/−* GCs at DIV21 and found a significant increase (1.94 ± 0.04 spines/μm) compared to WT controls (1.47 ± 0.05 spines/μm). This increase in *Plxna2−/−* GC spine density is similar to that observed in DIV21 *Sema5A−/−* GCs (2.23 ± 0.08 spines/μm) (*Figure 5D,E*, *Table 1*). Further, transfection of WT GCs with full-length *Sema5A* leads to a significant decrease in spine density (1.13 ± 0.05 spines/μm) (*Figures 3B and 5E*). However, *Plxna2−/−* GCs are resistant to the reduction in spine density caused by Sema5A overexpression (1.88 ± 0.05 spines/μm), showing that *Plxna2* is indeed required for Sema5A-mediated regulation of GC spine density (*Figure 5D,E*). Since we quantified spine density only in isolated prox1+ neurons that overexpress Sema5A and do not contact any nearby Sema5A-overexpressing neurons, these observations strongly suggest that Sema5A functions in *cis* to regulate dendritic spine density. To directly show that the PlexA2 cytoplasmic domain is required to inhibit GC spine density, *Plxna2−/−* cultures were transfected with full-length PlexA2 or PlexA2^{Δcyt}, which lacks the cytoplasmic domain. Though expression of full-length PlexA2 is sufficient to reduce spine density (1.38 ± 0.10 spines/μm) in *Plxna2−/−* GCs, no significant reduction was observed following transfection with PlexA2^{Δcyt} (2.03 ± 0.09 spines/μm) (*Figure 5D,E*).

We next addressed the role played by the PlexA2 RasGAP domain in the regulation of dendritic spine density since this plexin cytoplasmic domain has been shown to regulate the cytoskeleton during semaphorin signaling (*Rohm et al., 2000*; *He et al., 2009*; *Pasterkamp, 2012*). We generated a point-mutation in the PlexA2 cytoplasmic domain that abolishes its RasGAP activity (PlexA2$^{RR}$) but not Sema5A-Fc binding when expressed in COS7 cells. The PlexA2 rasGAP activity is required for the 'collapse' of COS7 cells (*Figure 5—figure supplement 2*). In primary GCs, PlexA2$^{RR}$ fails to rescue the spine phenotype observed in *Plxna2−/−*; Prox1+ GCs (1.97 ± 0.05 spines/μm) (*Figure 5D,E*), suggesting that the RasGAP activity directly participates in PlexA2-mediated regulation of dendritic spine density.

Transmembrane semaphorin–plexin receptor interactions in *cis* regulate neuronal repulsion in mouse sympathetic neurons, synaptic tiling in *Caenorhabditis elegans*, and starburst amacrine cell morphology and function (*Haklai-Topper et al., 2010*; *Mizumoto and Shen, 2013*; *Sun et al., 2013*). To better define Sema5A-PlexA2 signaling, we co-expressed PlexA2 with full-length Sema5A or membrane-bound Sema5A deletion mutants lacking either the cytoplasmic domain (S5A^{Δcyt}) or the 7TSRs (S5A^{ΔTSR})

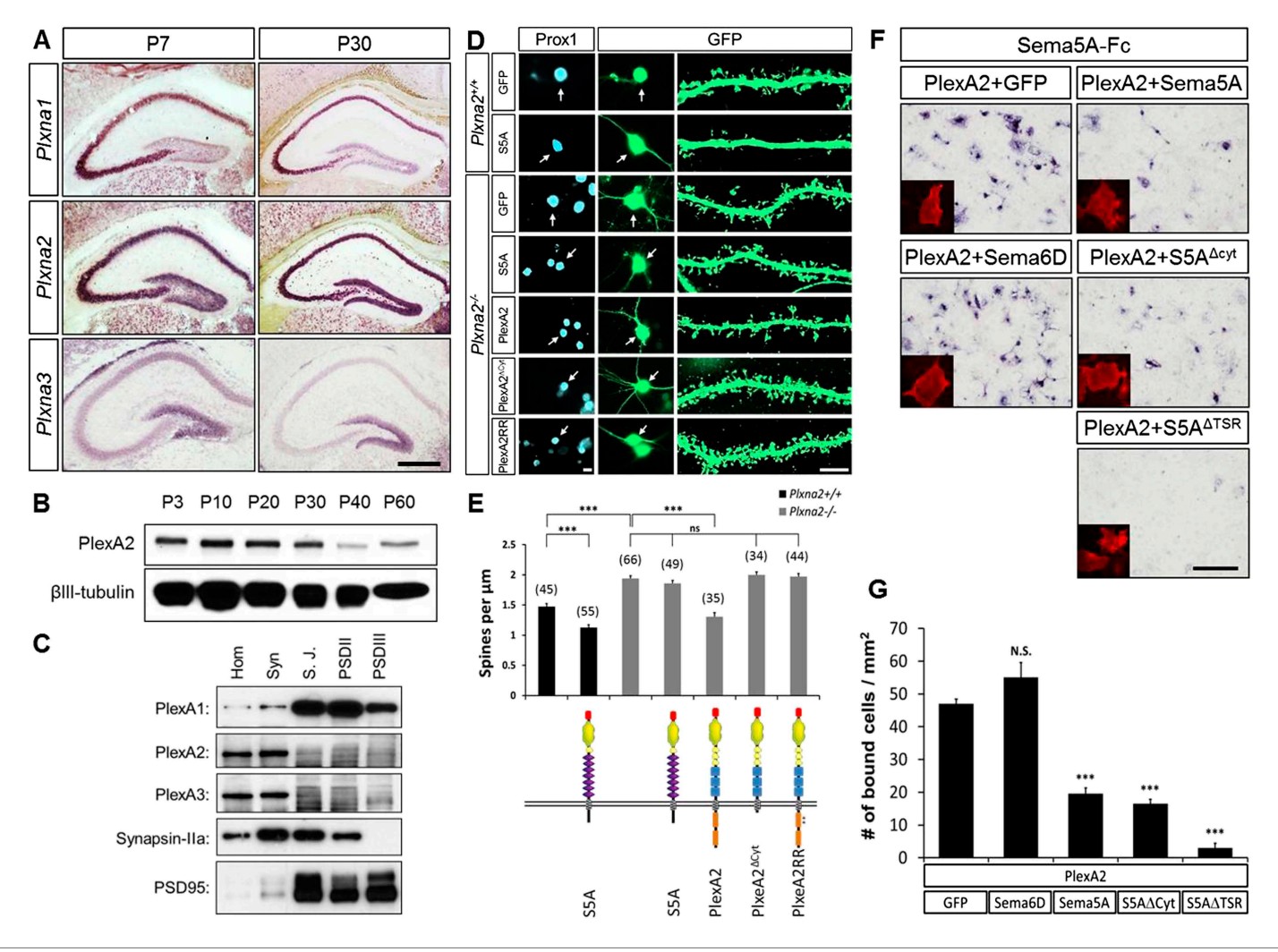

**Figure 5**. In dentate GCs, PlexA2 is necessary for Sema5A-mediated regulation of dendritic spine density. (**A**) Coronal sections of P7 and P30 mouse hippocampus labeled with riboprobes specific for *Plxna1*, *Plxna2*, and *Plxna3* transcripts. In the DG, strong expression of *Plxna2* is observed, while *Plxna1* is virtually absent. *Plxna3* mRNA expression is largely confined to more immature GCs located near the inner surface of the GCL. (**B**) Developmental time course of PlexA2 protein expression in the mouse hippocampus between P3 and P60, normalized to βIII-tubulin. (**C**) Distribution of PlexA1, PlexA2, and PlexA3 in synaptic density fractions prepared from P23 mouse hippocampus. The three PlexA family members are present in the synaptosomal (syn) fraction. PlexA1, but not PlexA2 or PlexA3, is enriched in the synaptic junction (S.J.) and postsynaptic density (PSD) fractions. As controls, markers enriched in postsynaptic fractions (anti-PSD95) and presynaptic fractions (anti-Synapsin-IIa) are shown. n = 4 independent experiments. (**D**) Representative images of DIV21 GCs from *Plxna2*[+/+] and *Plxna2*[−/−] mice transfected with GFP alone or GFP plus myc-tagged FL-Sema5A (S5A), FL-PlexA2 (PlexA2), PlexA2[Δcyt], or PlexA2[RR] expression constructs. Cultures were fixed and stained with anti-Prox1 (blue) and anti-GFP (green). The cell bodies of GFP-expressing GCs are labeled with an arrow and a dendritic segment of the same neuron is shown. (**E**) Quantification of spine density of GCs shown in panel **D**. WT neurons (black bars) and *Plxna2*[−/−] mutant neurons (grey bars). Numbers in brackets indicate GCs analyzed per condition. Values are represented as mean ± SEM from three animals for each condition. (**F**) *Cis* interaction between Sema5A and PlexinA2. Membrane-bound Sema5A presented in *cis*, and soluble Sema5A-Fc presented in *trans*, compete for binding to PlexinA2. Sema5A-Fc binding to COS7 cells transiently transfected to co-express PlexA2 and GFP, PlexA2 and Sema6D, PlexA2 and Sema5A, PlexA2 and Sema5A lacking the cytoplasmic domain (S5A[Δcyt]), or PlexA2 and Sema5A lacking the seven thrombospondin type-1 domains (S5A[ΔTSR]). Prior to binding, Sema5A-Fc was super-clustered with anti-Fc conjugated to human placental alkaline phosphatase (***Robak et al., 2009***). Surface expression of recombinant PlexA2 was confirmed in a parallel experiment by immunofluorescence labeling of its N-terminal myc epitope (small inserts). (**G**) Quantification of Sema5A-Fc bound to COS7 cells co-expressing PlexA2 together with GFP, Sema6D, Sema5A, S5A[Δcyt], or S5A[ΔTSR]. *Cis* expression of Sema5A constructs that harbor the sema domain significantly attenuates the binding of bath applied Sema5A-Fc to PlexA2. Error bars represent SEM from three independent experiments. ***indicates p < 0.0001. Two-tailed unpaired Student's *t* test. Scale bar in **A** = 50 μm.

*Figure 5. Continued on next page*

*Figure 5. Continued*

The following figure supplements are available for figure 5:

**Figure supplement 1**. Sema5A and PlexA2 are co-expressed in GCs and interact in *cis* and in *trans*.

**Figure supplement 2**. Sema5A-Fc–induced collapse of PlexA1- and PlexA2- expressing COS7 cells requires PlexA rasGAP activity.

**Figure supplement 3**. Molecular basis of Sema5A cis interaction with PlexA1 and PlexA2.

in COS7 cells. Co-expression of PlexA2 with Sema5A constructs that contain the Sema5A sema domain significantly attenuates binding of bath applied Sema5A-Fc (*Figure 5F,G*). In low-density HEK293T cell cultures, co-expression of individual PlexA family members with membrane-bound full-length Sema5A shows that PlexA1 and PlexA2, but not PlexA3, are part of a Sema5A complex that can be immunoprecipitated from cell lysates (*Figures 5—figure supplement 3A*). A membrane-bound Sema5A deletion construct lacking the 7 TSRs (S5A$^{\Delta TSR}$) interacts with full-length PlexA1 and PlexA2 when co-expressed with these receptors, but not PlexA3. The cytoplasmic domain of PlexA1 and PlexA2 are dispensable for this interaction with Sema5A (*Figures 5—figure supplement 3B*). Together, these observations strongly suggest that the sema-domain of Sema5A forms a complex with PlexA2 when presented in *cis* or in *trans*.

### *Plxna2*$^{-/-}$ mice show increased dentate GC synapse density

To examine the in vivo roles of PlexA1 and PlexA2 in GCs, we generated *Plxna1*$^{-/-}$;*Thy1-eGFP* and *Plxna2*$^{-/-}$;*Thy1-eGFP* mice and quantified spine density in these mutants at P33. Compared to WT mice (1.70 ± 0.04 spines/μm), we observed a significant increase in spine density in *Plxna2*$^{-/-}$ (2.24 ± 0.07 spines/μm), but not in *Plxna1*$^{-/-}$ (1.79 ± 0.05 spines/μm), mutants (*Figure 6A and B*, *Table 1*). This increase in GC spine density in *Plxna2*$^{-/-}$ mutants is comparable to that observed in *Sema5A*$^{-/-}$ mutants (2.05 ± 0.05 spines/μm). To analyze synaptic profiles at the ultra-structural level, and also to obtain a direct comparison between the *Sema5A*$^{-/-}$ and *Plxna2*$^{-/-}$ GC phenotypes, we performed scanning electron microscopy (sEM) followed by three-dimensional (3D) reconstructions of GC dendritic segments in 1-month-old WT, *Plxna2*$^{-/-}$ and *Sema5A*$^{-/-}$ mice. We analyzed dendrites in the outer one-third of the supragranular ML (*Figure 6—figure supplement 1A–I*), as we did with *Plxna2*$^{-/-}$;*Thy1eGFP* and *Sema5A*$^{-/-}$;*Thy1eGFP* mice, using immunofluorescence microscopy (*Figures 1E and 6A*). sEM revealed a robust supernumerary GC spine phenotype in *Plxna2*$^{-/-}$ mice (2.44 ± 0.19 spines/μm) and *Sema5A*$^{-/-}$ mice (2.65 ± 0.20 spines/μm) compared to WT (1.75 ± 0.11 spines/μm) (*Figure 6C,D*). Most dendritic spines reconstructed from WT (89.8%), *Plxna2*$^{-/-}$ (92.3%) and *Sema5A*$^{-/-}$ (93.9%) mice had a clearly identifiable PSD, a synaptic cleft and an opposed presynaptic bouton with vesicles (*Figure 6—figure supplement 1G,H,I*). The density of spines with no identifiable PSD was comparable among WT, *Sema5A*$^{-/-}$, or *Plxna2*$^{-/-}$ GCs (*Figure 6— figure supplement 1J*), and the mostly thin and filopodia-like structure of spines in these mutants suggests that they had not yet developed into mature synapses. No significant differences in spine head volume or PSD area were detected among WT, *Plxna2*$^{-/-}$, or *Sema5A*$^{-/-}$ mutants (*Figures 6— figure supplement 1K,L*). We found no evidence of dendritic spines that are contacted by more than one presynaptic bouton (*Figure 6—figure supplement 1M*). The increase in synapse density in *Plxna2* mice detected using sEM was independently confirmed with 2D EM (*Figures 6—figure supplement 1N–P*). These experiments show that synapse density in the GC outer ML is significantly increased in *Plxna2*$^{-/-}$ and *Sema5A*$^{-/-}$ mice, and they also show that Sema5A and PlexA2 negatively regulate excitatory synapse density in vivo.

### *Sema5A* and *Plxna2* genetically interact

We next assessed GC dendritic spine density in P33 heterozygous *Sema5A*$^{+/-}$ (1.65 ± 0.06 spines/μm) and *Plxna2*$^{+/-}$ (1.86 ± 0.05 spines/μm) mice, comparing these observations to compound heterozygous *Sema5A*$^{+/-}$;*Plxna2*$^{+/-}$ mice (2.39 ± 0.05 spines/μm) (*Figure 6E,F*). Only compound heterozygous mice show a significant increase in spine density, and this increase is comparable to that observed in *Sema5A*$^{-/-}$ and *PlexA2*$^{-/-}$ single mutants (*Table 1*). Therefore, *Sema5A* and *Plxna2* genetically interact, suggesting they function in the same signaling pathway in vivo. Together with our observations of

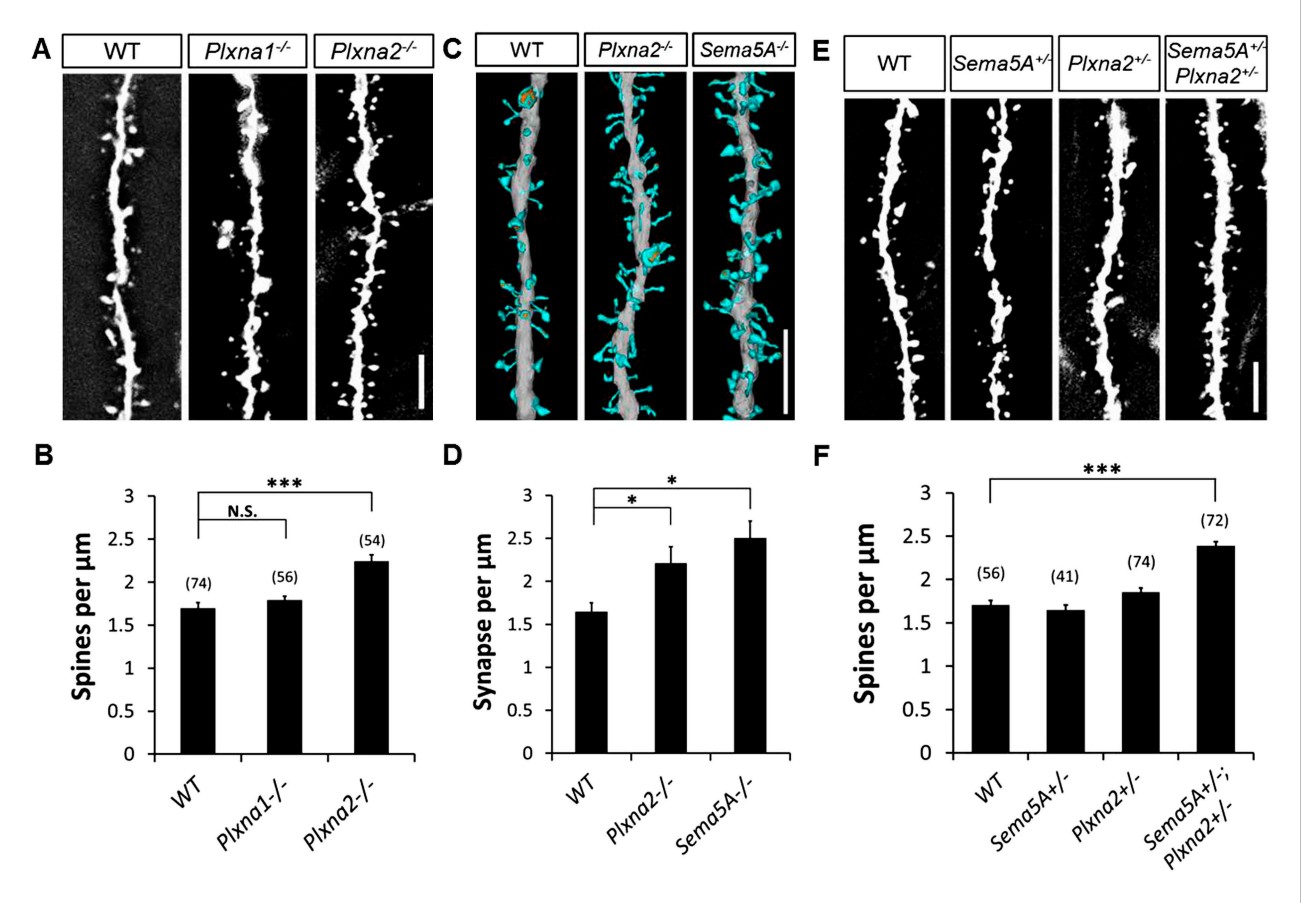

**Figure 6**. *Sema5A* and *Plxna2* genetically interact, and *Plxna2⁻/⁻* and *Sema5A⁻/⁻* mice show increased synapse density in dentate GCs. (**A**) Representative images of outer dendritic segments of GCs from P30 *Thy1-eGFPm* WT, *Plxna1⁻/⁻* and *Plxna2⁻/⁻* hippocampal sections. (**B**) Quantification of GC dendritic spine density reveals a significant increase in *Plxna2⁻/⁻* mice compared to *Plxna1⁻/⁻* and WT controls. Numbers in brackets indicate the dendritic segments analyzed per condition. Values are represented as mean ± SEM from three to four animals of each genotype. ***p < 0.001, two-tailed unpaired Student's *t* test. (**C**) Three-dimensional reconstruction of GC dendritic segments from P32 WT, *Plxna2⁻/⁻* and *Sema5A⁻/⁻* mice. The dendritic shaft is colored in gray, dendritic spines in blue and the PSDs in orange. A total of 110, 208, and 165 spines were reconstructed and analyzed for WT, *Plxna2⁻/⁻* and *Sema5A⁻/⁻* mice, respectively (n = 1 animal per genotype). (**D**) Quantification of spine density. Values are represented as mean ± SEM from three to four dendritic segments (each 20 to 22 μm in length) per genotype, *p < 0.05, two-tailed unpaired Student's *t* test. (**E**) Representative images of GC dendritic segments of P30 *Thy1-eGFPm* WT, *Sema5A⁺/⁻*, *Plxna2⁺/⁻*, and compound heterozygous *Sema5A⁺/⁻*; *Plxna2⁺/⁻* mice. (**F**) Quantification of GC spine densities of genotypes in **E** reveals significantly more spines in compound heterozygotes than WT and single heterozygotes. Numbers in brackets indicate the number of dendrites analyzed per condition. Values are represented as mean ± SEM from four animals per genotype. ***p < 0.0001, two-tailed unpaired Student's *t* test. Scale bars in **A**, **C** and **D** = 5 μm.

The following figure supplements are available for figure 6:

**Figure supplement 1**. Ultra-structural analysis of the *Plxna2⁻/⁻* and *Sema5A⁻/⁻* dentate gyrus molecular layer (ML).

**Figure supplement 2**. *Sema5A^LacZ/LacZ* mice have normal GC dendritic spine density.

direct Sema5A and PlexA2 binding, co-expression in GCs, localization to synaptosomal fractions, and function in primary GCs to regulate dendritic spine density, these data demonstrate that PlexA2 is a functional receptor for Sema5A, mediating Sema5A constraint of GC excitatory synapse density in vivo.

### *Sema5A⁻/⁻* mice exhibit impaired social behavior

A human genome-wide association study identified a significant association between a SNP near the *SEMA5A* locus and autism (*Weiss et al., 2009*). However, behavioral studies with a *Sema5A* gene-trap insertion mouse mutant (*Sema5A^lacZ/lacZ*) on a mixed C57Bl/6/129P2 background showed no behavioral

differences compared to C57Bl/6 control mice (*Gunn et al., 2011*). This gene-trap insertion is in the last coding exon of the *Sema5A* gene, generating a deletion mutant lacking the C-terminal portion of the cytoplasmic domain, a region of Sema5A that we find is not necessary for the regulation of dendritic spine density (*Figure 3B*). Indeed, when we analyzed GC spine density in these mice we did not observe a significant difference compared to WT controls, suggesting this *Sema5A* allele is not a null, or strong loss of function, mutation (*Figure 6—figure supplement 2*). Therefore, we extensively backcrossed our *Sema5A*⁻/⁻ mice to C57Bl/6 mice. These congenic *Sema5A*⁻/⁻ mutants were confirmed to be >99% C57Bl/6 and were subjected to a panel of behavioral assays. *Sema5A*⁻/⁻ congenic mutants show normal locomotor behavior in the open field test (*Figure 7—figure supplement 1A*), and mutant and WT mice exhibit similar anxiety levels (*Figures 7—figure supplement 1B,C*). They also spend a similar percentage of time in the open and closed arms of the elevated plus maze (*Figure 7—figure supplement 1D*). Contextual and cued fear conditioning revealed no difference between *Sema5A* WT and null mice (*Figure 7A,B*). In the three-chambered apparatus (*Yang et al., 2011*), a simple social approach task, both C57Bl/6 WT and *Sema5A*⁻/⁻ mice spent significantly more time engaged in nose-to-nose interactions with a mouse than sniffing an inanimate object (*Figure 7C*), a behavior representing normal sociability (*Roullet and Crawley, 2011*). We next assessed the time of direct nose-to-nose interaction of WT C57Bl/6 and *Sema5A*⁻/⁻ mice with a stranger vs a familiar mouse. WT mice show a strong preference for nose-to-nose contact with the stranger mouse, a behavior representing normal social interaction (*Yang et al., 2011*). This preference was not observed in *Sema5A*⁻/⁻ mice (*Figure 7D*), indicating that *Sema5A*⁻/⁻ mice exhibit deficits in sociability. These behavioral studies establish an endophenotype in *Sema5A*⁻/⁻ mice commonly associated with ASD.

## Discussion

We demonstrate here that Sema5A is a novel inhibitor of excitatory synapse formation and AMPA-type synaptic transmission. In the mouse hippocampus, Sema5A negatively regulates dendritic spine density in both developmentally born and adult-born dentate GCs. Loss of Sema5A-PlexA2 signaling results in the formation of supernumerary dendritic spines, exuberant excitatory synapses, and an increase in amplitude and frequency of excitatory synaptic transmission. Sema5A and PlexA2 are co-expressed in GCs and can both function cell-autonomously to prevent the formation of exuberant spine synapses. The rasGAP activity in the PlexA2 cytoplasmic domain is required for Sema5A-mediated inhibition of dendritic spine density. *Sema5A*⁻/⁻ congenic mice show normal locomotion and no evidence for elevated fear or anxiety. However, the loss of *Sema5A* leads to deficiencies in sociability. Collectively, these results suggest that intradendritic semaphorin–plexin interactions inhibit synapse density and synaptic transmission, and when perturbed, the absence of this signaling pathway leads to impaired social behavior, a hallmark of ASD.

### *Sema5A* negatively regulates GC synaptic strength

Developmentally born and adult-born GCs mature at different rates and in different environments, yet the sequence of neuronal maturation events is stereotyped and temporally segregated (*Esposito et al., 2005*). GC intrinsic programs may be advantageous in order to preserve the sequence of cellular maturation. Sema5A is strongly expressed in GCs during development and throughout adulthood. Similar to developmentally born GCs, we find that adult-born GCs employ Sema5A to negatively regulate dendritic spine density. In adult-born *Sema5A*⁻/⁻ GCs 3-weeks postmitosis, dendritic spine density is increased and correlates with an increase in mEPSC frequency. In addition, the loss of *Sema5A* increases the amplitude of AMPA receptor currents in these adult-born GCs. Because of its postsynaptic localization, Sema5A may normally negatively regulate AMPA receptor surface expression, synaptic distribution, or downstream signaling events that limit mEPSC amplitude. In primary hippocampal neurons, we find that the loss of *Sema5A* does not lead to an increase in total GluA1 expression or localization to synapses (data not shown). Sema5A associates with HSPGs and CSPGs (*Kantor et al., 2004*), and proteoglycans are known to participate in the regulation of surface GluA1 distribution through inhibition of AMPA receptor lateral diffusion at synapses. In the absence of Sema5A, the barrier function of CSPGs may be increased, resulting in elevated synaptic strength (*Allen and Barres, 2005*; *Frischknecht et al., 2009*; *Orlando et al., 2012*). More indirect mechanisms may also apply here, since two recent studies show that presynaptic HSPGs, including members of the glypican and syndecan families, interact with the LRRTM4 protein to promote the formation of excitatory synapses on GCs (*Siddiqui et al., 2013*; *de Wit et al., 2013*). Sema5A can interact with neural syndecans

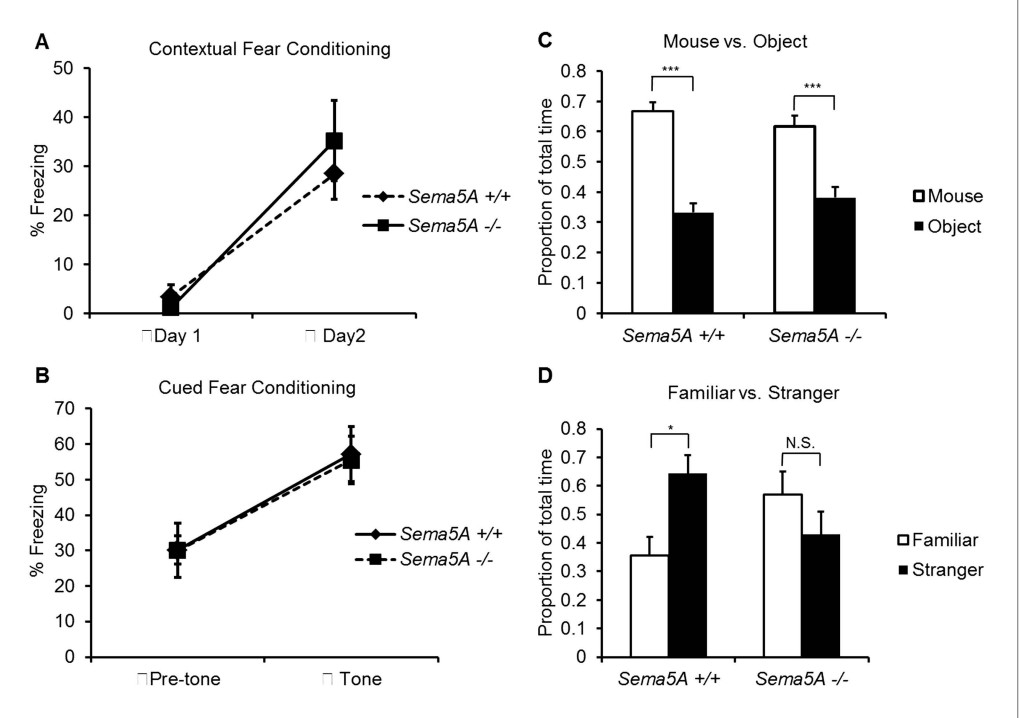

**Figure 7**. *Sema5A⁻/⁻* mice show abnormal social interaction. (**A** and **B**) Fear conditioning. *Sema5A⁺/⁺* and *Sema5A⁻/⁻* mice show no difference in freezing behavior under contextual fear conditioning (**A**) and cued fear conditioning (**B**) paradigms. Number of animals tested: n = 6 *Sema5A⁺/⁺* mice and n = 8 *Sema5A⁻/⁻* mice. Error bars represent ± SEM. (**C** and **D**) Three-chambered social interaction test: (**C**) given the choice, both *Sema5A⁺/⁺* and *Sema5A⁻/⁻* mice spent significantly more time engaged in nose-to-nose interaction with a mouse than sniffing an inanimate object. (**D**) *Sema5A⁺/⁺* mice spent significantly more time engaged in nose-to-nose interaction with a stranger mouse compared to a familiar mouse, whereas *Sema5A⁻/⁻* mice showed no preference. Number of animals tested: n = 8 *Sema5A⁺/⁺* mice and n = 8 *Sema5A⁻/⁻* mice for mouse vs inanimate object, and n = 7 *Sema5A⁺/⁺* mice and n = 7 *Sema5A⁻/⁻* mice for familiar vs stranger mouse. Error bars represent ± SEM. n.s. not significant, ***p < 0.001 and *p < 0.05 by two-tailed Student's *t* test.

The following figure supplement is available for figure 7:

**Figure supplement 1**. *Sema5A⁻/⁻* mice show normal anxiety levels and locomotor activity.

(*Kantor et al., 2004*), and at GC synapses Sema5A may compete with the synaptogenic activity of LRRTM4. While the recombinant sema domain of Sema5A is sufficient to 'rescue' the increased dendritic spine density observed in primary GCs null for *Sema5A⁻/⁻* in vitro, TSR-dependent proteoglycan interactions with endogenous Sema5A may influence the strength of sema-domain dependent *cis*- and *trans*-complexes in vivo. It is interesting in this context that, similar to what we observe in *Sema5A⁻/⁻* mice, loss of *LRRTM4* selectivity influences the density of excitatory synapses but has no effect on inhibitory synapse density. Additional studies will elucidate the precise mechanisms by which Sema5A negatively regulates synaptic strength, and whether it serves to directly counter synaptogenic pathways.

## PlexA2 is a novel Sema5A receptor in dentate GCs

In non-neuronal cells, Sema5A signals in a PlexB3-dependent manner to elicit distinct cytoskeletal responses, including collapse of NIH3T3 cells, motility of glioma cells, and migration of primary endothelial cells (*Artigiani et al., 2004*; *Li and Lee, 2010*). Neurite outgrowth in primary retinal neurons is inhibited by Sema5A and Sema5B, and this response is abrogated in *Plxna1*, *Plxna3* double null neurons (*Oster et al., 2003*; *Goldberg et al., 2004*; *Matsuoka et al., 2011*). Here, we identify PlexA2 as a high-affinity receptor for Sema5A in dentate GCs. Previous studies showed that Sema6A regulates GC axonal targeting in a *Plxna2*- and *Plxna4*-dependent manner (*Suto et al., 2007*), and also that the secreted molecule Sema3F signals through neuropilin-2/PlexA3 to control dendritic spine shape

and density in developing GCs (*Tran et al., 2009*). Similar to Sema3F, Sema5A regulates GC spine density, however Sema3F, but not Sema5A, apparently regulates spine shape. Sema3F is a secreted molecule, and thus can exert long-range effects, while Sema5A is a transmembrane molecule and acts locally. Sema5A, but not Sema3F, continues to be expressed in the adult dentate, and we show that Sema5A regulates synaptic density and function in adult-born GCs. These studies highlight the intricacies of semaphorin–plexin interactions with respect to the regulation of hippocampal circuit assembly at the subcellular level.

### Cell-autonomous Sema5A/PlexA2 signaling regulates GC synapse density

Many *cis* interactions between axon guidance molecules and their receptors, including ephrins/Ephs and Sema6s/PlexAs, attenuate receptor function (*Haklai-Topper et al., 2010*; *Yaron and Sprinzak, 2012*; *Sun et al., 2013*). *Cis* regulation of a membrane-bound semaphorin, *Sema1*, by a plexin receptor, *Plxn-1*, cell-autonomously controls the position of *C. elegans* synaptic boutons in developing DA9 motoneurons. *Sema1* and *Plxn-1* function to restrict synaptic bouton formation to appropriate subaxonal segments (*Mizumoto and Shen, 2013*). Similarly, the *cis*-interaction between PlexA2 and Sema5A on GC dendrites we describe here may suppress the formation of supernumerary protrusions that mature into excitatory synapses. In addition to this *cis*-interaction, Sema5A can also bind to PlexA2 when presented in *trans*. In cultured GCs this *trans*-interaction induces rapid dendritic spine elimination. In vivo, PlexA2-Sema5A *cis*-interactions may predominate since both molecules can be detected on the surface of cultured GC neurites. However, *trans* interactions may occur when filopodia or dendritic spines of two neighboring GCs compete for space on the same presynaptic bouton to form a multi-synapse bouton (*Toni et al., 2007*), or when different dendrites from the same GC neuron come into contact. Thus, our observations suggest that a combination of Sema5A-PlexA2 *cis* and *trans* interactions may ultimately determine GC spine density in vivo.

### *Sema5A* and autism

Genome-wide ASD association analyses have identified an intergenic SNP near the *SEMA5A* locus (*Weiss et al., 2009*), and this same study showed that *SEMA5A* expression is reduced in Brodmann area 19 of autism brains. An independent study showed that expression of *SEMA5A* is down regulated in transformed lymphocytes of autistic subjects (*Melin et al., 2006*). We used *Sema5A* null mice for behavioral testing, including the three-chambered apparatus, which provides a simple social approach task. *Sema5A*$^{-/-}$ mice exhibit impaired sociability, a core trait of ASD that has been observed in several mouse models of autism (*Roullet and Crawley, 2011*). Although altered social behavior may be secondary to increased anxiety, this appears unlikely since *Sema5A*$^{-/-}$ mice show no signs of anxiety in the open field test or in the elevated plus maze compared to WT controls. Because Sema5A is expressed broadly in the CNS, including in neuronal and non-neuronal cell types (*Zhang et al., 2014*), additional studies are warranted to determine the cellular basis of the behavioral deficits we observed in *Sema5A*$^{-/-}$ mice. Studies utilizing *Sema5A* conditional mutants that lack Sema5A in specific neural cell-types or specific brain structures are required to identify where *Sema5A* function is required for proper neural circuit development and to pin point which of these regions participate in normal social interaction behaviors.

Linking the synaptic defects observed in *Sema5A* mutants to the observed behavioral phenotypes is difficult. However, a growing number of ASD-associated genes encode proteins that function at the synapse, suggesting that altered synapse assembly and density is causally linked to ASD (*Peca and Feng, 2012*; *Voineagu and Eapen, 2013*). Human *SEMA5A* has been identified as an ASD susceptibility gene (*Weiss et al., 2009*), and we show here that its mouse homologue, *Sema5A*, functions as a negative regulator of synaptic transmission and formation of spiny synapses. When coupled with the impaired social behavior we observe in *Sema5A*$^{-/-}$ mice, our studies begin to provide mechanistic insight into how altered *SEMA5A* function may lead to brain illness, including autism.

## Materials and methods

### Mice

All mice used in this study were housed and cared for in accordance with NIH guidelines, and all research conducted was done with the approval of the University of Michigan Medical School and The Johns Hopkins University Committees on Use and Care of Animals. *Sema5A*$^{-/-}$, *Sema5A*$^{(flox/flox)}$,

*Sema5B⁻/⁻*, *Plxna1⁻/⁻*, *Plxna2⁻/⁻*, *Plxna3⁻/⁻*, and *Thy1-GFPm* mice were previously described (*Feng et al., 2000*; *Cheng et al., 2001*; *Yoshida et al., 2006*; *Matsuoka et al., 2011*). *Sema5A^{tm1Dgen}/J* mice express nuclear β-galactosidase (β-gal) under the control of the *Sema5A* promoter (*Sema5A^{lacZ}*) (*Gunn et al., 2011*) and were purchased from Jackson Labs. For behavioral studies, *Sema5A* mice were backcrossed onto a C57Bl/6 background using the speed congenic services provided by Charles River. Male mice used for behavioral studies were >99% congenic, as assessed by single nucleotide polymorphism (SNP) analysis (Charles River).

## DNA constructs

Rat *Sema5A* cDNA (Accession: NM_001107659) was used to generate expression constructs for Sema5A, including 6xMyc N-terminal-tagged full-length (FL)-Sema5A comprised of amino acid residues 23–1074 (23–1074 aa), membrane-bound Sema5A lacking the seven TSRs (6xMyc-Sema5A^{ΔTSR}; 23–539 and 951–1074 aa) and Sema5A lacking the cytoplasmic domain (6xMyc-Sema5A^{Δcyt}; 23–999 aa). Constructs were obtained by PCR amplification of the corresponding *Sema5A* sequences and subsequently cloned into the pCAG expression vector in frame with the Ig kappa signal sequence and the N-terminal 6xMyc tag. The following Fc fusion constructs were assembled in pcDNA3.0 (Invitrogen, Grand Island, NY): Sema5A ectodomain (Sema5A-Fc; 1–961 aa), sema domain (Sema5A^{Sema}-Fc; 1–539 aa), and seven TSR repeats (Sema5A^{TSR}-Fc; 536–960 aa). The expression and affinity purification of Fc fusion proteins, including NgR^{OMNI}-Fc, were carried out as described (*Robak et al., 2009*). Fusion proteins with C-terminal green fluorescent protein (GFP) in pcDNA3.0 included 6xMyc-FL-Sema5A-GFP and 6xMyc-Sema5A^{ΔTSR}-GFP. Mouse PlexA constructs with an N-terminal 6xMyc tag were generated by PCR cloning and included FL PlexA1, PlexA2, PlexA3, and PlexA4. The ectodomains of PlexA1 (PlexA1-Fc; 28–1237 aa), PlexA2 (PlexA2-Fc; 35–1232 aa), and the sema domain of PlexA1 (PlexA1^{SD}-Fc; 28–580 aa) were fused to the Fc portion of human IgG1. The PlexA2^{SD}-Fc and PlexA4^{SD}-Fc constructs were generously provided by F Suto (*Suto et al., 2007*). To obtain deletion mutants that lack the cytoplasmic domain, the ectodomain plus transmembrane spanning domain of PlexA1 (28–1283 aa) and PlexA2 (35–1280) were cloned into the pCAG expression vector. The PlexA1 RasGAP-deficient mutant was generously provided by A Puschel (*Rohm et al., 2000*). The PlexA2 RasGAP-deficient mutant (PlexA2^{AA}) was constructed by mutating two arginines, R1428A and R1429A, corresponding to key residues in the RasGAP catalytic domain of PlexA1 (*Rohm et al., 2000*).

## Histological procedures

P18 and P30 *Sema5A^{tm1Dgen}/J* mice were perfused transcardially with 4% paraformaldehyde (PFA). The brains were then dissected, post-fixed in 4% PFA for 30 min, cryoprotected in 30% sucrose in phosphate buffered saline pH7.4 (PBS) and flash frozen in dry-ice cooled isopentane. Horizontal sections were cut at 30-μm thickness, post-fixed in 4% PFA for 15 min, rinsed twice in 1× PBS, incubated with X-Gal solution (5 mM potassium ferricyanide, 5 mM potassium ferrocyanide, 2 mM MgCl₂, and 1 mg/ml 5-bromo-4-chloro-3-indolyl-b-D-galactopyranoside [X-Gal] in PBS) and developed overnight at 37°C. Alternatively, β-galactosidase expression in brain sections was visualized by anti-β-galactosidase (1:500, Abcam, Cambridge, MA) immunohistochemistry. In situ hybrdization with digoxigenin-labeled riboprobes specific for *Sema5A, Sema5B, Plxna1, Plxna2* and *Plxna3* was carried out described previously (*Matsuoka et al., 2011*).

For immunohistochemistry, coronal brain sections, cryosectioned at 25 μm and lifted on Superfrost⁺ microscope slides, were fixed in 4% PFA for 15 min and rinsed twice with 1× PBS for 5 min each. The sections were blocked with 5% horse serum, 0.3% Triton X-100 in PBS for 1 hr at room temperature, and incubation overnight at 4°C with the following primary antibodies: rabbit anti-Calbindin (1:1000, Swant, Switzerland), goat anti-Calretinin (1:1000, Swant), goat anti-Doublecortin (1:200, Santa Cruz), rabbit anti-GFP antibody (1:1000, Life Science). The sections were rinsed times times with 1× PBS for 5 min each, and incubated with Alexa Fluoro-conjugated secondary antibodies for 1 hr at room temperature. The sections were imaged and analyzed with an Olympus IX71 inverted fluorescence microscope.

Dendritic spine quantification: to quantify spine density, *Sema5A, Sema5B, Plxna1,* and *Plxna2* mice were crossed with *Thy1-GFPm* mice to genetically label a subset of postnatal hippocampal neurons. At P30–P33, mice were perfused transcardially with ice-cold 4% PFA in PBS and brains were postfixed for 2 hr in perfusion solution and cryoprotected in 30% sucrose in PBS. Coronal brain sections (8 μm thick) at the level of the rostral hippocampus were mounted on Superfrost⁺ microscope slides and fixed in 4% PFA for 15 min and processed for fluorescent microscopy. For imaging of GFP-labeled

neurons, z-stacks at 0.3-µm intervals were taken using the Deltavision RT imaging system (Applied Precision), equipped with an Olympus IX-70 inverted microscope with a 100× Oil objective plus an extra 1.6× zoom. Captured images were processed with 3D deconvolution to reduce background signals. For GCs, dendritic spine density in the outer one-third of the dorsal ML was quantified. For CA1 pyramidal neurons, spine density on the first and second order branches was quantified separately. For retrovirus-mediated birth-dating of adult-born GC, virus was injected into the DG of 5- to 6-weeks-old mice (*Ge et al., 2006*). 3 weeks later, brains were dissected, cryosectioned coronally at 40-µm thickness, and stained with anti-GFP antibody. Images were acquired with a 40× oil objective plus 5× zoom on a Zeiss LSM 710 META multiphoton confocal system (Carl Zeiss, Thornwood, NY) using a multi-track configuration. Only GCs in the dorsal molecular layer of DG were imaged and dendritic spines quantified as described for *Thy1-GFPm* mice.

## 3D reconstruction of dendritic segments

For serial block-face scanning electron microscopy (SBF-SEM), P30 WT, *Plxna2*$^{-/-}$ and *Sema5A*$^{-/-}$ mice were perfused transcardially for 2 min with ice-cold PBS and then for 10 min with freshly prepared, ice-cold 4% PFA and 2.5% glutaraldehyde in 0.1 M Na$^+$-cacodylate buffer (pH 7.4). Brains were dissected, postfixed in perfusion solution, and sectioned coronally in perfusion solution at 400 µm using a Leica vibrotome (TV1200). Brain slices that included the rostral hippocampus were fixed overnight in perfusion solution at 4°C and then processed for SBF-SEM according to instructions by Renovo Neural Inc. (Cleveland, OH). Briefly, tissue sections were rinsed in 0.1 M Na cacodylate buffer, stained with 0.1% tannic acid for 30 min that was followed by the staining regime developed by Renovo. Tissue was then dehydrated in ethanol and infiltrated and embedded in EPON 812. Approximately, 300 serial electron micrographs at 75-nm thickness were cut and imaged at a resolution of 7 nm/pix at Renovo Neural Inc., using a sigma VP scanning electron microscope (Carl Zeiss Microscopy, Jena, Germany) and a 3view door (Gatan Inc., Pleasanton, CA). For 3D reconstruction and quantification of dendritic segments, dendritic spine density, spine volume, spine head volume, and PSD surface, these structures were manually traced in individual sections using the PSD Reconstruct software (SynapseWeb).

## Primary neuronal cell culture

Hippocampal cultures enriched for GCs were established from WT and mutant mouse pups (P1–P2) by microdissection of dentate gyri. Briefly, the hippocampus was dissected and the DG separated from CA3/CA1. Combined dentate gyri were trypsinized for 15 min at 37°C, rinsed twice in HBSS (Gibco, Grand Island, NY), and mechanically dissociated in neurobasal medium containing 2% B27 supplement, 2 mM Glutamax, 50 unit/ml Penicillin, and 50 µg/ml streptomycin (Invitrogen). Cells were rinsed once in HBSS and once in culture medium and plated onto glass coverslips (Carolina Biologicals) coated with 100 µg/ml Poly-D-Lysine (Sigma, St. Louis, MO) and 10 µg/ml laminin (Life Technologies), at a density of 300,000 cells per 18-mm coverslip in a 12-well plate. Using lipofectamine 2000 (LFA), DIV4 cultures were transfected for 4 hr with a mixture of 1 µg of eGFP-expressing construct and 2 µl LFA per well of a 12-well plate. At DIV21, cultures were fixed and immunostained for identification of GCs and quantification of dendritic spine density.

## Immunocytochemistry

Briefly, primary neurons were fixed with 4% PFA at DIV21 for 15 min at room temperature, and blocked in PBS with 1% horse serum and 0.1% Triton X-100 (PHT) for 1 hr. Cells were incubated with primary antibodies in PHT overnight at 4°C, followed by 1-hr incubation with fluorophore-conjugated secondary antibodies at room temperature. Primary antibodies used included: rabbit anti-Prox1 (1:5000, Millipore), mouse anti-myc (1:1000, Developmental Studies Hybridoma Bank at the University of Iowa), chicken anti-GFP (1:1000; AVES, Tigard, OR), goat anti-GFP (1:500; Rockland, Limerick, PA), guinea pig anti-vGlut1 (1:1000; Millipore), mouse anti-PSD95 (1:500; Millipore), rabbit anti-VGAT (1:1000; Synaptic Systems), and mouse anti-gephyrin (1:1000; Synaptic Systems). Species-specific and Alexa Fluor-conjugated secondary antibodies (Life Technologies) were used at 1:1000.

## Isolation of synaptic density fractions and Western blot analysis

Freshly dissected mouse brains were homogenized in ice-cold 4 mM HEPES pH 7.4, 0.32 M sucrose, and protease inhibitor cocktail (1:100, Roche). Brain homogenate was centrifuged at 2000×*g* for 10 min to obtain supernatant 1 (S1) and pellet 1 (P1). The S1 fraction was centrifuged at 37,000×*g* for 30 min resulting in S2 and P2. Then, P2 was resuspended in ice-cold 4 mM HEPES, pH 7.4 and

fractionated by centrifugation (82,500×*g* for 2 hr) on a discontinuous sucrose gradient of 4 mM HEPES in 0.85 M, 1.0 M, and 1.2 M sucrose. The synaptosomal fraction (syn) was harvested at the 1.0/1.2 M sucrose interface, centrifuged at 150,000×*g* for 30 min, and resuspended in 80 mM TrisHCl pH 7.8. To isolate PSD fractions, synaptosomes were incubated with 40 mM TrisHCl pH 8.0, 0.5% Triton-X100 for 15 min on ice and then centrifuged at 32,000×*g* for 20 min. The pellet was resuspended in 40 mM TrisHCl pH 8.0 to obtain PSD I. The synaptic junction (SJ or PSD I) fraction was incubated with 40 mM TrisHCl pH 8.0, 0.5% Triton-X for 15 min on ice and centrifuged at 201,800×*g* for 1 hr. The pellet was resuspended in 40 mM TrisHCl pH 8.0 with 0.3% SDS to give PSD II fraction. To isolate the PSD III fraction, PSD I was incubated with 40 mM TrisHCl pH 8.0, 3% sarkosyl for 15 min and centrifuged at 201,800×*g* for 1 hr. The pellet was resuspended in 40 mM TrisHCl pH 8.0 to yield PSD III. Protein concentrations were measured with the BCA assay (Pierce) and samples mixed with 2× Laemmli buffer. 5 µg of each fraction was used for Western blot analysis. Primary antibodies used included mouse anti-PSD95 (1:2000; Millipore), rabbit anti-synaptophysin (1:1000; Santa Cruz), rabbit anti-Sema5A (1:1000, [*Matsuoka et al., 2011*]), rabbit anti-Sema5B (1:250, [*Matsuoka et al., 2011*]), rabbit anti-PlexA1 (1:2000; gift from Yutaka Yoshida), rabbit anti-PlexA2 (1:2000; gift from Hajime Fujisawa), rabbit anti-PlexA3 (1:500; Abcam), and mouse anti-neuropilin-2 (1:1000; Cell Signaling), mouse anti-Myc (1:1000, University of Iowa). Species-specific secondary antibodies were used at 1:5000 (Jackson, West Grove, PA).

## Immunoprecipitation

HEK293T cells were transfected with various combinations of 6xMyc-tagged FL-Sema5 and PlexA expression constructs, the deletion constructs PlexA-Δcyto, Sema5AΔTSR, or Sema5A-GFP. 24 hr after transfection, cells were incubated in ice-cold lysis buffer containing 20 mM TrisHCl (pH 7.5), 150 mM NaCl, 1% CHAPS (wt/vol), 1% Triton X-100 (vol/vol), and protease inhibitor cocktail. Cell lysates were precleared with Protein G Plus/Protein A–Agarose beads (Calbiochem), and then incubated with mouse anti-GFP antibody (Invitrogen) overnight at 4°C followed by incubation with Protein G Plus/Protein A–Agarose beads for an additional 4 hr. After the pull-down by centrifugation, beads were rinsed three times in lysis buffer, and bound proteins were analyzed by Western blotting. To assay for direct ligand–receptor interactions, Fc-tagged Sema5A ectodomain protein or Human IgG (50 nM) was incubated with 6xMyc-tagged PlexA1 or PlexA2 ectodomain produced in HEK293T cells, pulled down with Protein G Plus/Protein A–Agarose beads, and were analyzed for the presence of plexins by immunoblotting.

## Fusion protein binding studies

For ligand–receptor interaction studies, binding of recombinant protein to transfected COS7 cells was carried out as previously described (*Robak et al., 2009*). Briefly, COS7 cells were plated onto poly-D-lysine-coated 24-well plates and cultured overnight before transfection with Lipofectamine 2000 (Life Technologies) and expression constructs for WT and mutant versions of PlexA or Sema5A. After 24–36 hr, cells were incubated for 75 min with Fc-tagged fusion proteins preclustered with anti-human IgG conjugated to human placental alkaline phosphatase (AP) in OptiMEM (Promega, Madison, WI). Unbound fusion protein was removed by extensive rinsing with OptiMEM. Cells were then fixed with 1% formaldehyde in 60% acetone, rinsed twice in HBHA (HBS supplemented with 0.05% horse serum, 0.5% NaN₃), and once in HBS (Cellgro). Endogenous phosphatases were heat-inactivated by incubation at 65°C for 70 min. Binding of fusion protein was visualized by developing the AP reaction with NBT/BCIP substrate (Sigma). For binding of Sema5A-Fc and PlexinA2$^{SD}$-Fc fusion proteins to non-transfected primary mouse hippocampal neurons (DIV5-7), the same protocol was followed; however, NeuroBasal medium was used instead of OptiMEM.

## Dendritic spine collapse assay

Primary neuronal cultures enriched for GCs were established and transfected with an eGFP expression construct at DIV4 as described above. At DIV21, live imaging was performed and pictures taken with a Deltavision-RT Live Cell Imaging System at 5% CO₂ and 37°C at the University of Michigan Imaging Laboratories (MIL). Briefly, neurons on glass coverslips were placed in the microscope incubation chamber with 0.5 ml prewarmed B27 growth medium. GFP transfected neurons were imaged with a 100× objective and pictures taken every 5 min for 10 min before ligand treatment and then for an additional 120 min. At 10 min, Sema5A-Fc or human IgG (Sigma) preclustered with anti-human IgG (Promega) was added to the culture medium at a final concentration of 10 µg/ml. 2 hr after ligand incubation, cultures were fixed and stained with anti-Prox1 to identify which of the imaged cells were GCs. For quantification of dendritic spine density, only prox1+ cells were included.

## Stereotaxic injection of viral vectors

For stereotaxic injection of Lentiviral vectors (LV)-eGFP or LV-synapsin-eGFP-IRES-Cre (*Bateup et al., 2013*) into the hippocampus of P15 *Sema5A*$^{(flox/-)}$ mice, an automated injector pump (Stoetling) connected to a 5-μl Hamilton syringe was used. Briefly, anesthesia was induced by inhalation of 4% isoflurane mixed with oxygen and maintained by 2% isoflurane during surgery. After the plane of anesthesia was reached, skin on top of the skull was cleaned three times with iodine and ethanol wipes alternately, and a 1-cm long skin incision was made near the midline. After exposure of skull bone, a small hole was drilled on each side of the midline (ML) with the following coordinates: ML = ±1.27 mm, anterior-posterior (AP) = −1.65 mm from bregma. A 30-gauge needle was slowly inserted into the brain at a dorsal–ventral depth (DV) of −2.25 mm from the skull surface. Then, 0.5 μl of high titer (500× concentrated) LV was injected at a speed of 0.5 μl/min. LV-eGFP was injected on the left side and LV-synapsin-eGFP-IRES-Cre on the right side. The injection needle was left in place for an additional 2 min after the injection was completed and then withdrawn slowly. The skin was closed using veterinary glue, and the animals were placed on a 37°C heating pad for recovery. After surgery, mice were monitored regularly for 14 days and killed at P29 to assess dendritic spine density of transduced dentate GCs.

## Labeling of adult born GCs

Engineered self-inactivating murine retroviruses with a GFP reporter under the control of ubiquitin promoter were used to label and birth-date the proliferating cells and their progeny as previously described (*Ge et al., 2006*). High titers of engineered retroviruses were produced by co-transfection of retroviral vectors and VSVG into HEK293gp cells followed by ultracentrifugation of viral supernatant. Young adult (5–6 weeks old) *Sema5A*$^{-/-}$ mice and their littermates housed under standard conditions were anaesthetized with ketamine/xylazine, and retroviruses were stereotaxically injected into the dentate gyrus at four sites (0.5 μl per site) with the following coordinates: AP = −2 mm from bregma; lateral from ML = ±1.5 mm; and DV = −2.5 mm.

## Electrophysiology

Adult mice housed under standard conditions were anaesthetized 19–21 days after retroviral injection and processed for slice preparation as previously described (*Ge et al., 2006*). The brains were quickly removed into the ice-cold solution (in mM: 110 choline chloride, 2.5 KCl, 1.3 KH$_2$PO$_4$, 25.0 NaHCO$_3$, 0.5 CaCl$_2$, 7 MgSO$_4$, 20 dextrose, 1.3 sodium L-ascorbate, 0.6 sodium pyruvate, 5.0 kynurenic acid). Slices (275 μm thick) were cut using a vibrotome (Leica VT1200S) and transferred to a chamber containing the external solution (in mM: 125.0 NaCl, 2.5 KCl, 1.3 KH$_2$PO$_4$, 1.3 MgSO$_4$, 25.0 NaHCO$_3$, 2 CaCl$_2$, 1.3 sodium L-ascorbate, 0.6 sodium pyruvate, 10 dextrose, pH 7.4, 320 mOsm), bubbled with 95% O$_2$/5% CO$_2$. Electrophysiological recordings were obtained at 32°C–34°C. GFP$^+$ cells within the granule cell layer were visualized by DIC and fluorescence microscopy. The whole-cell patch-clamp configuration was employed in voltage-clamp mode (V$_m$ = −65 mV). Microelectrodes (4–6 MΩ) were pulled from borosilicate glass capillaries and filled with the internal solution containing (in mM) 120 potassium gluconate, 15 KCl, 4 MgCl$_2$, 0.1 EGTA, 10.0 HEPES, 4 ATP (magnesium salt), 0.3 GTP (sodium salt), 7 phosphocreatine (pH 7.4, 300 mOsm). The recordings were done in the presence of bicuculline (10 μM) to block GABAergic currents and TTX (1 μM) to block action potentials. Data were collected using an Axon 200B amplifier and acquired with a DigiData 1322A (Axon Instruments) at 10 kHz. Series and input resistances were monitored and only those with changes of less than 20% during experiments were analyzed. The series resistance ranged from 10 to 30 MΩ and was uncompensated.

## Behavioral studies

All behavioral experiments were performed on 3- to 4-month-old male C57Bl/6 congenic *Sema5A*$^{-/-}$ mice and their WT littermates.

### Open field

The test was performed using the Photobeam Activity System (San Diego Instruments). Mice were placed individually in the chambers and were allowed to move freely for 30 min. The number of beam interruptions caused by both horizontal and vertical movement was recorded. Total number of beam breaks was used to assess general locomotor activity. Percentage of time spent in the center was used to assess the anxiety level. Rearing represents the total number of vertical movements and was used to assess exploratory behavior.

## Elevated plus maze

Elevated plus maze was used to assess anxiety levels. Mice were placed in the center of the elevated plus maze and were allowed to move freely in the maze for 5 min. The behavior of the mice was videotaped and analyzed by Any-maze video tracking system (Stoelting Co.).

## Fear conditioning

On day 1, mice were placed in a fear-conditioning chamber for a total of 300 s. A 2000 Hz tone lasting 20 s was delivered at 210 s, followed by a 2-s foot shock. For contextual fear conditioning, mice were returned to the same chamber on day 2 and allowed to move freely for a total of 300 s. No tone or foot shocks were delivered. Freezing behavior was recorded and analyzed by FreezeScan software (Celver Sys Inc.) at 10-s intervals. For cued fear conditioning, mice were placed in a different chamber in a different room on day 2 and allowed to move freely for a total of 300 s. White noise was present for the duration of the test. A 2000 Hz tone lasting 20 s was delivered at 210 s. Freezing behavior was recorded and analyzed by FreezeScan software (Celver Sys Inc.) at 10-s intervals.

## Social interaction studies

The three-chamber social interaction test was used as described previously (*Yang et al., 2011*). Briefly, the subject mouse was placed in the center chamber with nothing in the side chambers and allowed to move freely between chambers and acclimate for 5 min. Subsequently, a cage with a WT C57Bl/6 mouse and a cage with an inanimate object were placed in each of the side chambers. The subject mouse was allowed to enter each chamber freely and interact with the mouse or object for 5 min. Finally, the inanimate object was replaced with a new WT C57Bl/6 mouse, and the subject mouse was allowed to enter each chamber freely and interact with either mouse for 5 min. The behavior of the mice was videotaped and analyzed by Any-maze video tracking system (Stoelting Co.). Video of each subject mouse was reviewed and time engaged in nose-to-nose interaction with other mice and sniffing the inanimate object was quantified manually.

## Acknowledgements

We thank Yutaka Yoshida for providing us with the *Plxna1*[−/−], *Plxna2*[−/−], and *Plxna3*[−/−] mice, Fumikazu Suto for PlexA2[SD]-Fc and PlexA4[SD]-Fc constructs and anti-PlexA2 antiserum, and Andreas Puschel for the PlexA1(RasGAP-RR) construct. We thank Onanong Chivatakarn and Ryota Matsuoka for their help with the initial analysis of the *Sema5A* hippocampal formation, Martín Riccomagno for assistance with viral vector injections, Xiao-Feng Zhao for help with histochemical procedures, and Dontais Johnson for technical assistance. We thank Alcino Silva, Jack Parent, and members of the Kolodkin and Giger laboratories for critical reading of the manuscript. This work is supported by the Maryland Stem Cell Research Fund (JS and G-lM); NS048271 and HD069184 (G-lM); MH59199 (ALK); NS081281 (RJG); and the Dr Miriam and Sheldon Adelson Medical Research Foundation (G-lM and RJG). ALK is an investigator of the Howard Hughes Medical Institute.

## Additional information

### Funding

| Funder | Author |
| --- | --- |
| National Institute of Neurological Disorders and Stroke | Roman J Giger |
| National Institute of Mental Health | Alex L Kolodkin, Guo-li Ming |
| Howard Hughes Medical Institute | Alex L Kolodkin |
| Dr. Miriam and Sheldon G. Adelson Medical Research Foundation | Guo-li Ming, Roman J Giger |
| Maryland Stem Cell Research Fund | Juan Song, Guo-li Ming |

The funders had no role in study design, data collection and interpretation, or the decision to submit the work for publication.

## Author contributions

YD, S-HW, JS, YM, Acquisition of data, Analysis and interpretation of data; G-M, Conception and design; ALK, RJG, Conception and design, Analysis and interpretation of data, Drafting or revising the article

## Ethics

Animal experimentation: All mice used in this study were housed and cared for in accordance with NIH guidelines, and all research conducted was done with the approval of the University of Michigan Medical School (UCUCA protocols PRO00002466 and PRO00001645) and The Johns Hopkins University (MO14M50 and MO12M381) Committees on Use and Care of Animals. All surgery was performed under sodium pentobarbital anesthesia, and every effort was made to minimize suffering.

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
