## [Decision Letter]

Thank you for sending your work entitled “Semaphorin5A Inhibits Synaptogenesis in Early Postnatal- and Adult-born Hippocampal Dentate Granule Cells” for consideration at *eLife*. Your article has been favorably evaluated by a Senior editor, Freda Miller (Reviewing editor), and 2 reviewers, both of whom, Paul Frankland and Avraham Yaron, have agreed to reveal their identity.

The Reviewing editor and the two reviewers discussed their comments before we reached this decision, and the Reviewing editor has assembled the following comments to help you prepare a revised submission.

We all felt that this was a rigorous, interesting, and experimentally conclusive manuscript. In particular, this manuscript comprehensively characterized the role of semaphorin5A in synaptogenesis in early and late postnatally generated dentate gyrus granule cells. The experiments provided convincing evidence that loss of semaphorin5A leads to an increase in spine density and increased AMPA-mediated synaptic responses using a variety of detailed anatomical, cell biological and electrophysiological approaches. Moreover, their genetic experiments showed that loss of Plexin-A2 leads to a similar spine phenotype, and provided evidence for a genetic interaction. Since GWAS studies have implicated SEMA5A as an autism risk gene, it was exciting that the authors found that mice lacking this gene exhibit behavioral phenotypes relevant to autism (social interaction).

In summary, the reviewers felt this was an impressive series of experiments that would be of interest to a broad readership, given the fact that there are few known negative regulators of synaptogenesis, and the potential implications of these findings for autism. To strengthen the manuscript further, the reviewers felt that no new experiments were needed, but that a number of relatively minor revisions should be made, as follows.

Minor comments:

1) Figure 1. The authors write: “Low levels of Sema5B are detected in the PSDII fraction”, but the data also indicate that Sema5B is highly enriched in this fraction. As we do not see it in the other fractions, suggesting that it is highly specific to this neuronal sub-compartment, the authors should refer to this in the text.

2) Figure 4. Several things are not clear to me in this blot. 1. What are the bands that are detected in the Sema5A-FC blot where IgG is used (lanes 1, 3). 2. It seems that in this blot the interaction of Sema5A with Plexin-A2 is poor in comparison to the interaction with Plexin-A1. Maybe the authors have some quantitative data from additional blots? Or they can omit this experiment, as they do not refer to this specific piece of data in the main text.

3) Discussion: The authors discuss the role of HSPGs in connection with Sema5A, but the binding of Sema5A to HSPGs is mediated by the TSR domain, which is dispensable for the phenotypes that are described in this work.

4) Linking the spine phenotypes to the observed behavioral phenotypes is tremendously difficult to do, and is beyond the scope of the current paper. However, some discussion of the challenges and limitations seems warranted.

---

## [Author Response]

*1)*
Figure 1*. The authors write: “Low levels of Sema5B are detected in the PSDII fraction”, but the data also indicate that Sema5B is highly enriched in this fraction. As we do not see it in the other fractions, suggesting that it is highly specific to this neuronal sub-compartment, the authors should refer to this in the text*.

In our revised manuscript we have addressed this point and reworded the sentence as follows: “Sema5B is enriched postsynaptically and only found in the PSDIII fraction.”

*2)*
Figure 4*. Several things are not clear to me in this blot. 1. What are the bands that are detected in the Sema5A-FC blot where IgG is used (lanes 1, 3). 2. It seems that in this blot the interaction of Sema5A with Plexin-A2 is poor in comparison to the interaction with Plexin-A1. Maybe the authors have some quantitative data from additional blots? Or they can omit this experiment, as they do not refer to this specific piece of data in the main text*.

We have revised the figure legend for Figure 4 to clarify these issues. Here are the answerers to the reviewers’ questions: A) The bands detected in the Sema5A-Fc blot (second from top) are Sema5A-Fc (lanes 2 and 4) and control IgG in lanes 1 and 3.

B) The interactions of Sema5A-Fc with PlexinA1 and PlexinA2 are comparable in strength (we provide saturation binding curves in Figure 4–figure supplement C), the calculated affinities are between 10 and 13 nM.

3) Discussion: The authors discuss the role of HSPGs in connection with Sema5A, but the binding of Sema5A to HSPGs is mediated by the TSR domain, which is dispensable for the phenotypes that are described in this work

Agreed, in Figure 3 we show that overexpression of the recombinant sema-domain of Sema5A is sufficient to “rescue” the supernumerary spine phenotype observed in cultured *Sema5A*^*-/-*^ GCs. However, we think that the strength of the interaction of the sema-domain of endogenously expressed Sema5A may be regulated, at least in part, through proteoglycan interactions with the TSRs of Sema5A. To make this clearer, we have revised the Discussion accordingly:

“While the recombinant sema domain of Sema5A is sufficient to “rescue” the increased dendritic spine density observed in primary GCs null for *Sema5A*^*-/-*^, TSR-dependent proteoglycan interactions with endogenous Sema5A may influence the strength of sema-domain dependent *cis*- and *trans*-complexes in vivo.”

*4) Linking the spine phenotypes to the observed behavioral phenotypes is tremendously difficult to do, and is beyond the scope of the current paper. However, some discussion of the challenges and limitations seems warranted*.

We agree with the reviewer and have revised our Discussion accordingly: “Because Sema5A is expressed broadly in the CNS, including in neuronal and non-neuronal cell types [1], additional studies are warranted to determine the cellular basis of the behavioral deficits we observed in *Sema5A*^*-/-*^ mice. Studies utilizing *Sema5A* conditional mutants that lack Sema5A in specific neural cell-types or specific brain structures are required to identify where *Sema5A* function is required for proper neural circuit development and to pin point which of these regions participate in normal social interaction behaviors. Linking the synaptic defects observed in *Sema5A* mutants to the observed behavioral phenotypes is difficult. However, a growing number of ASD-associated genes encode proteins that function at the synapse, suggesting that altered synapse assembly and density is causally linked to ASD.”